# $f$-DOMAIN-ADVERSARIAL LEARNING: THEORY AND ALGORITHMS FOR UNSUPERVISED DOMAIN ADAPTATION WITH NEURAL NETWORKS

## ABSTRACT

The problem of unsupervised domain adaptation arises in a variety of practical applications where the distribution of the training samples differs from those used at test time. The existing theory of domain adaptation derived generalization bounds based on divergence measures that are hard to optimize in practice. This has led to a large disconnect between theory and state-of-the-art methods. In this paper, we propose a novel domain-adversarial framework that introduces new theory for domain adaptation and leads to practical learning algorithms with neural networks. In particular, we derive a novel generalization bound that utilizes a new measure of discrepancy between distributions based on a variational characterization of $f$-divergences. We show that our bound recovers the theoretical results from Ben-David et al. (2010a) as a special case with a particular choice of divergence, and also supports divergences typically used in practice. We derive a general algorithm for domain-adversarial learning for the complete family of $f$-divergences. We provide empirical results for several $f$-divergences and show that some, not considered previously in domain-adversarial learning, achieve state-of-the-art results in practice. We provide empirical insights into how choosing a particular divergence affects the transfer performance on real-world datasets. By further recognizing the optimization problem as a Stackelberg game, we utilize the latest optimizers from the game optimization literature, achieving additional performance boosts in our training algorithm. We show that our $f$-domain adversarial framework achieves state-of-the-art results on the challenging Office-31 and Office-Home datasets without extra hyperparameters.

## 1 INTRODUCTION

The ability to learn new concepts and skills from general-purpose data and transfer them to similar scenarios is critical in many modern applications. For example, it is often the case that the learner has access to only a small (unlabeled) subset of data on its domain of interest, but has access to a larger labeled dataset (for the same task) in a domain that is similar to the target domain. If the gap between these two domains is not considerable, we may expect to train a model by using the labeled and unlabeled data, and to generalize well to the target dataset. This scenario is called *unsupervised domain adaptation*, and it is the focus of this paper.

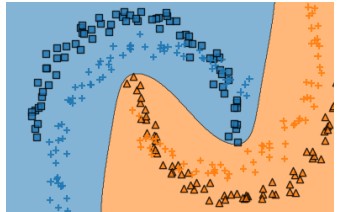

Figure 1: *Domain Adaptation*. A learner is trained on abundant labeled data and is expected to perform well in the target domain (marked as **+**). Decision boundaries correspond to a 2-layers neural net trained using $f$-DAL.

The paramount importance of domain adaptation (DA) has led to remarkable advances in the field. From a theoretical point of view, the seminal works of Ben-David et al. (2007; 2010a;b); Mansour et al. (2009) provided generalization bounds for unsupervised DA based on discrepancy measures that are a reduction of the Total Variation (TV). More recently, Zhang et al. (2019) took one step further and proposed the Margin Disparity Discrepancy (MDD) with the aim of closing the gap between theory and algorithms. Their notion of discrepancy is tailored to margin losses and builds on the observation of only taking a single supremum over the class set

to make optimization easier. Moreover, theories based on weighted combination of hypotheses for multiple source DA have also been developed (Hoffman et al., 2018a).

From an algorithmic perspective, specifically in the context of neural networks, Ganin & Lempitsky (2015); Ganin et al. (2016) proposed the idea of learning domain-invariant representations as a two-player zero-sum game. This approach led to a plethora of methods including state-of-the-art approaches such as Shu et al. (2018); Long et al. (2018); Hoffman et al. (2018b); Zhang et al. (2019). While these methods were explained with insights from the theory of Ben-David et al. (2010a), and more recently through MDD (Zhang et al., 2019), in deep neural networks, both the $\mathcal{H}\Delta\mathcal{H}$ divergence (Ben-David et al., 2010a) and MDD are hard to optimize, and *ad-hoc* objectives have been introduced to minimize the divergence between source and target distributions in a representation space. This has led to a disconnect between theory and the current SoTA methods. Specifically, approaches that follow Ganin et al. (2016) minimize a Jensen Shanon (JS) divergence, while the practical objective of MDD can be interpreted as minimizing a $\gamma$-weighted JS divergence. From the optimization perspective, the game-optimization nature of the problem has been ignored, and these min-max objectives are usually optimized using Gradient Descent Ascent (GDA) (referred to as Gradient Descent (GD) with the Gradient Reversal Layer (GRL)). Paradoxically, the *last iterate* of GDA is known to not converge to a Nash equilibrium even in simple bilinear games (Nemirovsky & Yudin, 1983).

The aim of this paper is to provide a novel perspective on the domain-adversarial problem by deriving theory that generalizes previous seminal works and translates into a new general framework that supports the complete family of $f$-divergences and is practical for modern neural networks. In particular, we introduce a novel measure of discrepancy between distributions and derive its corresponding learning bounds. Our notion of discrepancy is based on a variational characterization of $f$-divergences and includes both previous theoretical results (i.e. based on reductions of the TV) and practical results (i.e. based on JS). We empirically show that any $f$-divergence can be used to learn invariant representations. Most importantly, we show that several divergences that were not considered previously in domain-adversarial learning achieve SoTA results in practice. From an optimization point of view, we observe that under mild conditions, the optimal solution of our framework is a Stackelberg equilibrium. This allows us to plug-and-play the latest optimizers from the recent min-max optimization literature within our framework.

We also discuss practical considerations in deep networks, and compare how learning invariant representations for different choices of divergence affects the transfer performance on real-world datasets. We further discuss the practical gains (for popular $f$-divergences) that can be achieved by introducing more advanced optimizers. We will release code upon acceptance.

## 2 PRELIMINARIES

In this paper, we focus on the unsupervised domain adaptation scenario. During training, we assume that the learner has access to a source dataset of $n_s$ *labeled* examples $S = \{(x_i^s, y_i^s)\}_{i=1}^{n_s}$, and a target dataset of $n_t$ *unlabeled* examples $T = \{(x_i^t)\}_{i=1}^{n_t}$, where the source inputs $x_i^s$ are sampled i.i.d. from a distribution $P_s$ (source distribution) over the input space $\mathcal{X}$ and the target inputs $x_i^t$ are sampled i.i.d. from a distribution $P_t$ (target distribution) over $\mathcal{X}$. Usually, in the case of binary classification, we have $\mathcal{Y} = \{0, 1\}$ and in the multiclass classification scenario, $\mathcal{Y} = \{1, ..., k\}$. When $\mathcal{X}$ or $\mathcal{Y}$ cannot be inferred from the context or other assumptions are required, we will mention it explicitly.

We denote a labeling function by $f : \mathcal{X} \to \mathcal{Y}$, and the source and target labeling functions by $f_s$ and $f_t$, respectively. The task of unsupervised domain adaptation is to find a hypothesis function $h : \mathcal{X} \to \mathcal{Y}$ that generalizes to the target dataset T (i.e., to make as few errors as possible by comparing with the ground truth label $f_t(x_i^t)$). The risk of a hypothesis $h$ w.r.t. the labeling function $f$, using a loss function $\ell : \mathcal{Y} \times \mathcal{Y} \to \mathbb{R}_+$ under distribution $\mathcal{D}$ is defined as: $R_{\mathcal{D}}^\ell(h, f) := \mathbb{E}_{x \sim \mathcal{D}}[\ell(h(x), f(x))]$. We also assume that $\ell$ satisfies the triangle inequality. For simplicity of notation, we define $R_S^\ell(h) := R_{P_s}^\ell(h, f_s)$ and $R_T^\ell(h) := R_{P_t}^\ell(h, f_t)$ where the indices $S$ and $T$ refer to the source and target domains, respectively. We additionally use $\hat{R}_S, \hat{R}_T$ to refer to the empirical risks over the source dataset S and the target dataset T.

| Divergence | $\phi(x)$ | $\phi^*(t)$ | $\phi'(1)$ | $g(x)$ |
|---|---|---|---|---|
| Kullback-Leibler (KL) | $x \log x$ | $\exp(t-1)$ | $1$ | $x$ |
| Reverse KL (KL-rev) | $-\log x$ | $-1 - \log(-t)$ | $-1$ | $-\exp x$ |
| Jensen-Shannon (JS) | $-(x+1)\log \frac{1+x}{2} + x \log x$ | $-\log(2 - e^t)$ | $0$ | $\log \frac{2}{1+\exp(-x)}$ |
| Pearson $\chi^2$ | $(x-1)^2$ | $t^2/4 + t$ | $0$ | $x$ |
| Total Variation (TV) | $\frac{1}{2}|x-1|$ | $\mathbf{1}_{-1/2 \leq t \leq 1/2}$ | $[-1/2, 1/2]$ | $\frac{1}{2}\tanh x$ |

Table 1: Popular $f$-divergences, their conjugate functions and choices of $g$. We take $\hat{l}(a,b) = g(b_{\operatorname{argmax} a})$.

## 2.1 COMPARING SOURCE AND TARGET DOMAINS WITH $f$-DIVERGENCES

A key component of domain adaptation is the study of the discrepancy between the source and target distributions. This differentiates transductive approaches and more generally transfer learning from traditional supervised learning methods. In our work, we derive generalization bounds that capture the entire family of $f$-divergences. We define new discrepancies between source and target distributions, based on the variational characterization of popular choices of $f$-divergences. These new discrepancies play a fundamental role in our work.

**Definition 1** ($f$-**divergence,** Csiszár (**1967**); Ali & Silvey (**1966**))**.** *Let $P_s$ and $P_t$ two distributions functions with densities $p_s$ and $p_t$, respectively. Let $p_s$ and $p_t$ be absolute continuous with respect to a base measure $dx$. Let $\phi : \mathbb{R}_+ \to \mathbb{R}$ be a convex, lower semi-continuous function that satisfies $\phi(1) = 0$. The $f$-divergence $D_\phi$ is defined as:*

$$D_\phi(P_s||P_t) = \int p_t(x)\, \phi\left(\frac{p_s(x)}{p_t(x)}\right) dx. \tag{2.1}$$

**Variational characterization of $f$-divergences:** Nguyen et al. (2010) derive a general variational method that estimates $f$-divergences from samples by turning the estimation problem into variational optimization. They show that any $f$-divergence can be written as (see details in Appendix A.2):

$$D_\phi(P_s||P_t) \geq \sup_{T \in \mathcal{T}} \mathbb{E}_{x \sim P_s}[T(x)] - \mathbb{E}_{x \sim P_t}[\phi^*(T(x))] \tag{2.2}$$

where $\phi^*$ is the (Fenchel) conjugate function of $\phi : \mathbb{R}_+ \to \mathbb{R}$ defined as $\phi^*(y) := \sup_{x \in \mathbb{R}_+}\{xy - \phi(x)\}$, and $T : \mathcal{X} \to \operatorname{dom} \phi^*$. The equality holds if $\mathcal{T}$ is the set of all measurable functions. Many popular divergences that are heavily used in machine learning and information theory are special cases of $f$-divergences. We summarize them and their conjugate function in Table 1. For simplicity, we assume in the following that $\mathcal{X} \subseteq \mathbb{R}^n$ and each density (i.e $p_s$ and $p_t$) is absolutely continuous.

## 3 DOMAIN ADAPTATION THEORY

Domain adaptation approaches generally build upon the idea of bounding the gap between the source and target domains' error functions in terms of the discrepancy between their probability distributions. Measuring the similarity between the distributions $P_s$ and $P_t$ is thus critical in the derivation of generalization bounds and/or the design of algorithms. We remind the reader of the seminal work of Ben-David et al. (2010a) that bounds the risk of any binary classifier in the hypothesis class $\mathcal{H}$ with the following theorem:

**Theorem 1.** *If $\ell(x,y) = |h(x) - y|$ and $\mathcal{H}$ is a class of functions, then for any $h \in \mathcal{H}$ we have:*

$$R_T^\ell(h) \leq R_S^\ell(h) + D_{\mathrm{TV}}(P_s||P_t) + \min\{\mathbb{E}_{x \sim P_s}[|f_t(x) - f_s(x)|], \mathbb{E}_{x \sim P_t}[|f_t(x) - f_s(x)|]\}. \tag{3.1}$$

Here $D_{\mathrm{TV}}(P_s||P_t) := \sup_{T \in \mathcal{T}} |\mathbb{E}_{x \sim P_s}[T(x)] - \mathbb{E}_{x \sim P_t}[T(x)]|$ is the TV and $\mathcal{T}$ is the set of measurable functions. TV is an $f$-divergence such that $\phi(x) = |x - 1|$ in Definition 1. For any function $\phi(x) \geq |x - 1|$, one can replace $D_{\mathrm{TV}}(P_s||P_t)$ in eq. 3.1 with $D_\phi(P_s||P_t)$. Theorem 1 thus bounds a classifiers target error in terms of the source error, the divergence between the two domains, and the dissimilarity of the labeling functions. Unfortunately, $D_{\mathrm{TV}}(P_s||P_t)$ cannot be estimated from finite samples of arbitrary distributions (Kifer et al., 2004). It is also a very loose upper bound as it involves the supremum over all measurable functions and does not account for the hypothesis class.

### 3.1 Measuring discrepancy with $f$-divergences

This section introduces a new discrepancy that aims to solve the two aforementioned problems, namely **(1)** estimation of the divergence from finite samples of arbitrary distributions (Lemma 2) and **(2)** restriction of the discrepancy to the set including the hypothesis class $\mathcal{H}$. (Defs. 2 and 3). We show in Sec. 3.2 how these allows to extend the bounds studied in Ben-David et al. (2010a).

**Definition 2 ($D_{\mathcal{H}}^{\phi}$ discrepancy).** *Let $\phi^*$ be the Fenchel conjugate of a convex, lower semi-continuous function $\phi$ that satisfies $\phi(1) = 0$, and let $\hat{\mathcal{T}}$ be a set of measurable functions such that $\hat{\mathcal{T}} = \{\ell(h(x), h'(x)) : h, h' \in \mathcal{H}\}$. We define the discrepancy between $P_s$ and $P_t$ as:*

$$D_{\mathcal{H}}^{\phi}(P_s||P_t) := \sup_{h,h' \in \mathcal{H}} |\mathbb{E}_{x \sim P_s}[\ell(h(x), h'(x))] - \mathbb{E}_{x \sim P_t}[\phi^*(\ell(h(x), h'(x)))]|. \quad (3.2)$$

The $D_{\mathcal{H}}^{\phi}$ discrepancy can be interpreted as a lower bound estimator of a general class of $f$-divergences (Lemma 1). Therefore, for any hypothesis class $\mathcal{H}$ and choice of $\phi$, $D_{\mathcal{H}}^{\phi}$ is never larger than its corresponding $f$-divergence. We show in Lemma 2 that its computation can be bounded in terms of finite examples. Finally, we recover the $\mathcal{H}\Delta\mathcal{H}$ divergence (Ben-David et al., 2010a) if we consider $\phi^*(t) = t$ and $\ell(h(x), h'(x)) = \mathbf{1}[h(x) \neq h'(x)]$, which corresponds to the TV.

**Definition 3 ($D_{h,\mathcal{H}}^{\phi}$ discrepancy).** *Suppose the same conditions as above, the discrepancy between two distributions $P_s$ and $P_t$ is defined by:*

$$D_{h,\mathcal{H}}^{\phi}(P_s||P_t) := \sup_{h' \in \mathcal{H}} |\mathbb{E}_{x \sim P_s}[\ell(h(x), h'(x))] - \mathbb{E}_{x \sim P_t}[\phi^*(\ell(h(x), h'(x)))]|. \quad (3.3)$$

Taking the supremum of $D_{h,\mathcal{H}}^{\phi}$ over $h \in \mathcal{H}$, we obtain $D_{\mathcal{H}}^{\phi}$, and thus $D_{h,\mathcal{H}}^{\phi}(P_s||P_t) \leq D_{\mathcal{H}}^{\phi}(P_s||P_t)$. This bound will be useful when deriving practical algorithms.

**Lemma 1 (lower bound).** *For any two functions $h, h'$ in $\mathcal{H}$, we have:*

$$|R_S^{\ell}(h, h') - R_T^{\phi^* \circ \ell}(h, h')| \leq D_{h,\mathcal{H}}^{\phi}(P_s||P_t) \leq D_{\mathcal{H}}^{\phi}(P_s||P_t) \leq D_{\phi}(P_s||P_t). \quad (3.4)$$

Lemma 1 is fundamental in the derivation of divergence-based generalization bounds for DA. Specifically, it bounds the gap between the source and target domains' error functions in terms of the discrepancy between their distributions using $f$-divergences. We now show that the $D_{h,\mathcal{H}}^{\phi}$ can be estimated from finite samples.

**Lemma 2.** *Suppose $\ell : \mathcal{Y} \times \mathcal{Y} \to [0, 1]$, $\phi^*$ L-Lipschitz, and $[0, 1] \subset \operatorname{dom} \phi^*$. Let S and T be two empirical distributions corresponding to datasets containing n datapoints sampled i.i.d. from $P_s$ and $P_t$, respectively. Let us note $\mathfrak{R}$ the Rademacher complexity of a given class of functions, and $\ell \circ \mathcal{H} := \{x \mapsto \ell(h(x), h'(x)) : h, h' \in \mathcal{H}\}$. $\forall \delta \in (0, 1)$, we have with probability of at least $1 - \delta$:*

$$|D_{h,\mathcal{H}}^{\phi}(P_s||P_t) - D_{h,\mathcal{H}}^{\phi}(S||T)| \leq 2\mathfrak{R}_{P_s}(\ell \circ \mathcal{H}) + 2L\mathfrak{R}_{P_t}(\ell \circ \mathcal{H}) + 2\sqrt{(-\log \delta)/(2n)}. \quad (3.5)$$

In Lemma 2, we show that the empirical $D_{h,\mathcal{H}}^{\phi}$ converges to the true $D_{h,\mathcal{H}}^{\phi}$ discrepancy. It can then be estimated using a set of finite samples from the two distributions. The gap is bounded by the complexity of the hypothesis class and the number of examples ($n$). This result will be also important in the derivation of Theorem 3.

### 3.2 Domain Adaptation: Generalization Bounds

We now provide a novel generalization bound to estimate the error of a classifier in the target domain using the proposed $D_{h,\mathcal{H}}^{\phi}$ divergence and results from the previous section. We also provide a generalization Rademacher complexity bound for a binary classifier based on the estimation of the $D_{h,\mathcal{H}}^{\phi}$ from finite samples. We show that our bound generalizes previous existing results in Appendix D.1.

**Theorem 2 (generalization bound).** *Suppose $\ell : \mathcal{Y} \times \mathcal{Y} \to [0, 1] \subset \operatorname{dom} \phi^*$ and that $\ell(a, b) \leq \ell(a, c) + \ell(c, b)$ for any $a, b, c \in \mathcal{Y}$. Denote $\lambda^* := R_S^{\ell}(h^*) + R_T^{\ell}(h^*)$, and let $h^*$ be the ideal joint hypothesis. We have:*

$$R_T^{\ell}(h) \leq R_S^{\ell}(h) + D_{h,\mathcal{H}}^{\phi}(P_s||P_t) + \lambda^*. \quad (3.6)$$

The three terms in this upper bound share similarity with the bounds proposed by Ben-David et al. (2010a) and more recently by Zhang et al. (2019). The main difference lies in the discrepancy being used to compare the two marginal distributions. In the case of Ben-David et al. (2010a), they use the $\mathcal{H}\Delta\mathcal{H}$ divergence (a reduction of the TV), and in Zhang et al. (2019), they use the MDD. In our case, we use a reduction of a lower bound estimator of a variational characterization of the general $f$-divergences. This generalizes the TV (and thus Ben-David et al. (2010a)) and also includes popular divergences typically used in practice (see Appendix D). Intuitively, the first term in the bound accounts for the source error, the second corresponds to the discrepancy between the marginal distributions, and the third measures the ideal joint hypothesis ($\lambda^*$). If $\mathcal{H}$ is expressive enough and the labeling functions are similar, this last term could be reduced to a small value. The ideal joint hypothesis incorporates the notion of adaptability: when the optimal hypothesis performs poorly in either domain, we cannot expect successful adaptation.

**Theorem 3 (generalization bound with Rademacher complexity).** *Let $\ell : \mathcal{Y} \times \mathcal{Y} \to [0,1]$ and $\phi^*$ be L-Lipschitz. Let S and T be two empirical distributions (i.e. datasets containing $n$ data points sampled i.i.d. from $P_s$ and $P_t$, respectively). Denote $\hat{\lambda}_\phi^* := \hat{R}_S^\ell(h^*) + \hat{R}_T^\ell(h^*)$. $\forall \delta \in (0,1)$, we have with probability of at least $1 - \delta$:*

$$R_T^\ell(h) \leq \hat{R}_S^\ell(h) + D_{h,\mathcal{H}}^\phi(S\|T) + \hat{\lambda}_\phi^*$$
$$+ 6\Re_S(\ell \circ \mathcal{H}) + 2(1 + L)\Re_T(\ell \circ \mathcal{H}) + 5\sqrt{(-\log \delta)/(2n)}. \qquad (3.7)$$

In Theorem 3, we show the computation of our generalization bound for a binary classifier in terms of the Rademacher complexity of the class $\mathcal{H}$. We see that under the assumption of an ideal joint hypothesis, $\hat{\lambda}_\phi^*$, the generalization error can be reduced by jointly minimizing the risk in the source domain, the discrepancy between the two distributions, and regularizing the model to limit the complexity of the hypothesis class. We take all these into account when deriving practical algorithms in the next sections.

# 4 $f$-DOMAIN ADVERSARIAL LEARNING ($f$-DAL)

We now use the theory presented in the previous sections to derive a novel generalized domain adversarial learning framework. The key idea of domain-adversarial training is to simultaneously minimize the source error and align the two distributions in a representation space $\mathcal{Z}$. Specifically, we let a hypothesis $h$ be the composition of $h = \hat{h} \circ g$ (i.e let $\mathcal{H} := \{\hat{h} \circ g : \hat{h} \in \hat{\mathcal{H}}, g \in \mathcal{G}\}$ with $\hat{\mathcal{H}}$ another function class) where $g : \mathcal{X} \to \mathcal{Z}$. This can be interpreted as a mapping that pushes forward the two densities $p_s$ and $p_t$ to a representation space $\mathcal{Z}$ where a classifier $\hat{h} \in \hat{\mathcal{H}}$ operates. Consequently, we refer to $p_s^z := g\#p_s$ and $p_t^z := g\#p_t$ as the push-forwards of the source and target domain densities, respectively. Figure 2 illustrates the $f$-DAL framework.

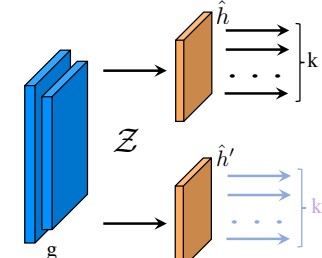

Figure 2: **$f$-DAL framework.** We interpret $h : \mathcal{X} \to \mathcal{Y}$ as the composition of two nets $h = \hat{h} \circ g$, where $g : \mathcal{X} \to \mathcal{Z}$ and $\hat{h}$ is a classifier that operates in a representation space $\mathcal{Z}$. Inspired by the theory, we let $\hat{h}'$ be another net of the same topology than $\hat{h}$. This is intuitively interpreted as a per-category domain classifier. Our framework is different from domain-adversarial frameworks that follows from (Ganin et al., 2016) since they use a global domain-classifier or discriminator.

Clearly from Theorem 2, for adaptation to be possible in the representation space $\mathcal{Z}$, there has to be an $\hat{h} \in \hat{\mathcal{H}}$, such that the ideal joint risk $\lambda^*$ is negligible. This condition is necessary even if $p_s^z = p_t^z$. In other words, we need the difference between $p_s^z$ and $p_t^z$ to be small, and the ideal joint risk $\lambda^*$ to be negligible. These are both sufficient and necessary conditions. We refer the reader to Ben-David et al. (2010b) for details on the impossibility theorems for DA. Consequently, we state the following:

**Assumption 1.** *There is a $g \in \mathcal{G}$ and $\hat{h}^* \in \hat{\mathcal{H}}$, such that the ideal joint risk ($\lambda^*$) is negligible. We also assume that the class-conditional distributions [1] between source and target are similar.*

While these assumptions may seem restrictive, they are ubiquitous in modern DA methods, including SoTA methods i.e Ganin et al. (2016); Long et al. (2018); Hoffman et al. (2018b); Zhang et al. (2019)

---

[1]Also referred to as the coovariate-shift assumption (Shimodaira, 2000). The source and target domains only differ in their marginals according to the input space.

(sometimes not explicitly mentioned). Moreover, neural networks are generally known to be able to learn rich and powerful representations, and in practical scenarios, $g$ and $\hat{h}$ are both neural networks.

From Theorem 2 and Assumption 1, the target risk $R_T^\ell(h)$ can be optimized by jointly minimizing the error in the source domain and the discrepancy between the two distributions. Letting $y := f_s(x)$, an optimization objective can be clearly written as:

$$\min_{\hat{h} \in \hat{\mathcal{H}}} \mathbb{E}_{z \sim p_s^z}[\ell(\hat{h}(z), y)] + D_{\hat{h}, \hat{\mathcal{H}}}^\phi(p_s^z || p_t^z). \tag{4.1}$$

Here, $\ell$ is a surrogate loss function used to minimize the empirical risk in the source domain. Nonetheless, it does not have to be the binary classification loss (i.e it can be the cross-entropy loss). Under some assumptions (Proposition 1) and the use of Lemma 1, the minimization problem in equation 4.1 can be upper bounded (hence replaced) by the following min-max objective[2]:

$$\min_{\hat{h} \in \hat{\mathcal{H}}} \max_{\hat{h}' \in \hat{\mathcal{H}}} \mathbb{E}_{z \sim p_s^z}[\ell(\hat{h}(z), y)] + \underbrace{\mathbb{E}_{z \sim p_s^z}[\hat{\ell}(\hat{h}'(z), \hat{h}(z))] - \mathbb{E}_{z \sim p_t^z}[(\phi^* \circ \hat{\ell})(\hat{h}'(z), \hat{h}(z))]}_{d_{s,t}} \tag{4.2}$$

where we refer to the difference between the last two terms as $d_{s,t}$. We now formalize this result.

**Proposition 1.** *Suppose $d_{s,t}$ takes the form shown in equation 4.2 with $\hat{\ell}(\hat{h}'(z), \hat{h}(z)) \to \mathrm{dom}\, \phi^*$ and that for any $\hat{h} \in \hat{\mathcal{H}}$, there exists $\hat{h}' \in \hat{\mathcal{H}}$ s.t. $\hat{\ell}(\hat{h}'(z), \hat{h}(z)) = \phi'(\frac{p_s^z(z)}{p_t^z(z)})$ for any $z \in \mathrm{supp}(p_t^z(z))$, with $\phi'$ the derivative of $\phi$. The optimal $d_{s,t}$ is $D_\phi(P_s^z || P_t^z)$ (i.e $\max_{\hat{h}' \in \hat{\mathcal{H}}} d_{s,t} = D_\phi(P_s^z || P_t^z)$).*

If we let the feature extractor $g \in \mathcal{G}$ be the one that minimizes both the source error and the discrepancy term, equation 4.2 can be rewritten as:

$$\min_{\hat{h} \in \hat{\mathcal{H}}, g \in \mathcal{G}} \max_{\hat{h}' \in \hat{\mathcal{H}}} \mathbb{E}_{x \sim p_s}[\ell(\hat{h} \circ g, y)] + \mathbb{E}_{x \sim p_s}[\hat{\ell}(\hat{h}' \circ g, \hat{h} \circ g)] - \mathbb{E}_{x \sim p_t}[(\phi^* \circ \hat{\ell})(\hat{h}' \circ g, \hat{h} \circ g)]. \tag{4.3}$$

The choice of $\hat{\ell}$ is "*somewhat arbitrary*" as stated for GANs in Nowozin et al. (2016). For the multiclass scenario, we let $\hat{\ell}(a, b) = g(b_{\mathrm{argmax}\, a})$, where argmax $a$ is the index of the largest element of vector $a$. For the binary case, we define $\hat{\ell}(\_, b) = g(b)$. This implies that we choose the domain of $\hat{\ell}$ to be $\mathbb{R}^k \times \mathbb{R}^k$ with $k$ categories for the multi-class scenario and $\mathbb{R}$ for binary classification. Intuitively, $\hat{h}'$ is an auxiliary per-category domain classifier. *Note that this is different from Ganin et al. (2016) where there is a unique domain classifier or discriminator.* For the choice of $g$, we follow Nowozin et al. (2016) and choose it to be a monotonically increasing function when possible. We summarize our choices of $\hat{\ell}$ for different $f$-divergences in Table 1. We also show other choices of $\hat{\ell}$ including generalizations of previous methods in Appendix D.

$\gamma$-**weighted JS divergence.** If we relax the need for $\phi(1) = 0$ in Proposition 1, the new objective only shifts by a constant, e.g, $\max_{\hat{h}' \in \hat{\mathcal{H}}} d_{s,t} = D_{\hat{\phi}}(P_s^z || P_t^z) + \phi(1)$ with $\hat{\phi}(x) := \phi(x) - \phi(1)$. By Lemma 4 (Appendix D), we can then rescale $\phi^*$, and $\phi$ will change accordingly. This allows to include the *practical objective* from Zhang et al. (2019) as part of our framework (i.e. $\gamma$-weighted JS, see Appendix D). While these can be done for the general family of divergences, we do not pursue this line of research in practice as it requires additional hyperparameter tuning of $\gamma$.

## 4.1 OPTIMALITY IN $f$-DAL

The main objective of our framework (i.e equation 4.3) is a minimax optimization problem and our desired (optimal) solution is under mild assumptions a Stackelberg equilibrium. This key observation allows us to incorporate in our framework the latest optimizers from the game-optimization literature. We now formalize and prove this concept. Based on this, we propose to use the extragradient algorithm and its aggressive version within our framework. We show the effectiveness through a toy example (Figure 3, Appendix E) and also empirically in our large scale experiments.

**Stackelberg equilibria in $f$-DAL.** We show the existence of Stackelberg equilibria in $f$-Domain Adversarial Learning ($f$-DAL). Let $\mathcal{G}$ and $\hat{\mathcal{H}}$ be a class of functions defined by a fixed parametric functional (i.e neural networks with fixed architecture), and define $\omega_1$ such that is a vector composed

---

[2]Indeed, under these assumptions, $d_{s,t}$ can be seen as an upper bound for the $D_{h,\mathcal{H}}^\phi$ discrepancy.

of the parameters of the feature extractor $g$ and the source classifier $\hat{h}$. Similarly, let $\omega_2$ be the parameters of the auxiliary classifier $\hat{h}'$, and $\Omega_1$ and $\Omega_2$ denote their separate domains. Equation (4.2) can be rewritten as:

$$\min_{\omega_1 \in \Omega_1} \max_{\omega_2 \in \Omega_2} V(\omega_1, \omega_2). \tag{4.4}$$

In general, $V$ is nonconvex in $\omega_1$ and nonconcave in $\omega_2$, and for the min-max game in equation 4.4, Nash equilibria may not exist as in (Farnia & Ozdaglar, 2020). A Stackelberg equilibrium is more general than Nash equilibrium (see definition in Definition 5) and reflects the sequential nature of our zero-sum game equation 4.4. We now show that the optimal solution of $f$-DAL is a Stackelberg equilibrium. Such an equilibrium is a stationary point under the assumption that $V(\omega_1, \cdot)$ is (locally) strongly concave in $\omega_2$ (Evtushenko, 1974), and we can then use gradient algorithms to search for such a desirable solution. In the following theorem, we use the explicit form of push-forward to emphasize the dependence on the feature extractor $g$, rather than $p_s^z, p_t^z$.

**Theorem 4 (Stackelberg equilibrium, informal).** *Suppose $d_{s,t}$ takes the form shown in equation 4.2, and assume that (a) There exists an optimal $g^* \in \mathcal{G}$ that maps both the source and the target distribution to the same distribution. (b) There exists an optimal classifier that yields the ground truth in a neighborhood. (c) For any $g \in \mathcal{G}$ and $\hat{h} \in \mathcal{H}$, there exists $\hat{h}'$ that achieves $\hat{\ell}(\hat{h}'(z), \hat{h}(z)) = \phi'((g\#p_s)(z)/(g\#p_t)(z))$. Then the objective of $f$-adversarial learning has a Stackelberg equilibrium at $(\hat{h}^*, g^*, \hat{h}'^*)$.*

When $\ell(\hat{h}, \hat{h}') = \ell(\hat{h}')$ (e.g., in a binary classification scenario or in Ganin et al. (2016)), the Stackelberg equilibrium can be shown to be a Nash equilibrium (see Theorem 6).

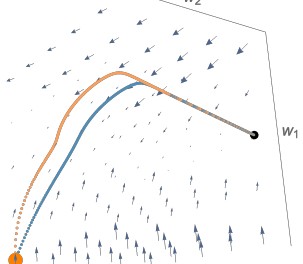

**Extra-gradient algorithms.** We have shown that the optimal solution in $f$-DAL is a Stackelberg equilibrium which is more general than a Nash equilibrium. For convergence to a Nash equilibrium, the simplest method is GDA. However, the last iterate of GDA does not converge in the bilinear case (e.g. Nemirovsky & Yudin, 1983). To accelerate and stabilize the convergence, the extra-gradient (EG) method was proposed in (Korpelevich, 1976) . It was recently shown (Zhang et al., 2020; Hsieh et al., 2020) that having an aggressive extra-step is even more stable than vanilla EG, and is more suitable for convergence to Stackelberg equilibria (Zhang et al., 2020). With the aim of quantifying whether exploiting Theorem 4 leads to practical gains, we follow those works, and therefore let the extrapolation step to be

Figure 3: Comparison of GDA vs AExG in a toy task with JS as divergence. $\mathcal{G}$ is the class of quadratic functions and $\hat{\mathcal{H}}$ is linear. AExG can accelerate the convergence to the optimal solution. (Appendix E)

larger. We refer to this algorithm as *Aggressive Extra-Gradient* (AExG). We illustrate this in a simple example (Appendix E), whose convergence/trajectories are shown in Figure 3 and we will explore AExG further in the experimental section.

## 5 EXPERIMENTAL RESULTS

We present experimental results of our framework in practical scenarios. In these scenarios, the learner is a neural network and the input domain is the set of natural images. Specifically, we aim to answer the following questions: **(1)** How does choosing a particular divergence affect the domain adaptation performance among different datasets? **(2)** Is there a better universal notion of $f$-divergence that achieves significant performance gains across different datasets and thus helps generalization? **(3)** Are there considerable practical gains by exploiting the fact that the optimal solution of $f$-DAL is a Stackelberg Equilibrium? **(4)** How does our theoretical framework compare in practice to existing SoTA methods? We also compare in Figure 4 the difference in interpretation of the auxiliary classifier of $f$-DAL vs Ganin et al. (2016). The comparison shows significant performance boost for the same divergence i.e DANN vs $f$-DAL (JS)

**Datasets.** In our experiments we use two main datasets. We use **(1)** the *Office-31* dataset (Saenko et al., 2010) which contains 4,652 images and 31 categories, collected from three distinct domains: Amazon **(A)**, Webcam **(W)** and DSLR **(D)**. We also use **(2)** the *Office-Home* dataset (Venkateswara et al., 2017). This is a more complex dataset containing 15,500 images from four different domains: Artistic images, Clip Art, Product images, and Real-world images. In each of our experiments, we report the average over 3 different seeds.

| Method | A → W | D → W | W → D | A → D | D → A | W → A | Avg |
|---|---|---|---|---|---|---|---|
| ResNet-50 (He et al., 2016) | 68.4±0.2 | 96.7±0.1 | 99.3±0.1 | 68.9±0.2 | 62.5±0.3 | 60.7±0.3 | 76.1 |
| DANN (Ganin et al., 2016) | 82.0±0.4 | 96.9±0.2 | 99.1±0.1 | 79.7±0.4 | 68.2±0.4 | 67.4±0.5 | 82.2 |
| ADDA (Tzeng et al., 2017) | 86.2±0.5 | 96.2±0.3 | 98.4±0.3 | 77.8±0.3 | 69.5±0.4 | 68.9±0.5 | 82.9 |
| JAN (Long et al., 2017) | 85.4±0.3 | 97.4±0.2 | 99.8±0.2 | 84.7±0.3 | 68.6±0.3 | 70.0±0.4 | 84.3 |
| GTA (Sankaranarayanan et al., 2018) | 89.5±0.5 | 97.9±0.3 | 99.8±0.4 | 87.7±0.5 | 72.8±0.3 | 71.4±0.4 | 86.5 |
| MCD (Saito et al., 2018) | 88.6±0.2 | 98.5±0.1 | **100.0**±.0 | 92.2±0.2 | 69.5±0.1 | 69.7±0.3 | 86.5 |
| CDAN Long et al. (2018) | 94.1±0.1 | 98.6±0.1 | **100.0**±.0 | 92.9±0.2 | 71.0±0.3 | 69.3±0.3 | 87.7 |
| MDD (Zhang et al., 2019) | 94.5±0.3 | 98.4±0.1 | **100.0**±.0 | 93.5±0.2 | **74.6**±0.3 | 72.2±0.1 | 88.9 |
| Ours (Pearson $\chi^2$, SGD) | **95.4** ±0.7 | 98.4 ± 0.2 | **100.0**±.0 | 93.8 ±0.4 | 73.5 ±1.1 | 74.2 ±0.5 | 89.2 |
| Ours (Pearson $\chi^2$, AExG) | 95.3 ±0.1 | **98.8** ± 0.3 | **100.0**±.0 | **94.2** ±0.6 | 73.3 ±0.3 | **75.3** ±0.2 | **89.5** |

Table 2: Comparison vs previous unsupervised domain adaptation approaches on the Office-31 benchmark. Accuracy represented in (%) with average and standard deviation. Impressively, our approach achieves SoTA results without the need of additional techniques (i.e CDAN) or additional hyperparameters (i.e MDD/$\gamma$-JS).

**Implementation Details**: We implement our algorithm in PyTorch. We use ResNet-50 (He et al., 2016) pretrained on ImageNet (Deng et al., 2009) as the feature extractor. The main classifier ($\hat{h}$) and auxiliary classifier ($\hat{h}'$) are both 2 layers neural nets with Leaky-Relu activation functions. We use spectral normalization (SN) as in Miyato et al. (2018) only for these two (i.e $\hat{h}$ and $\hat{h}'$). We did not see any transfer improvement by using it, neither by using Leaky-Relu activation functions instead of Relu. The reason for this was to avoid gradient issues and instabilities during training for some divergences (i.e KL, TV) in the first epochs. For simplicity, and fair comparison with previous work, we perform simultaneous updates using the GRL. We also use the GRL warm-up strategy. This is standard in most DA frameworks and follows from Eq. (14) in Ganin & Lempitsky (2015). For optimization, we use **1)** mini-batch (32) SGD (or GDA) with the Nesterov momentum (0.9). **2)** For experiments using AExG, we take inspiration from Gidel et al. (2019) and implement a version of the ExtraGradient with momentum (0.9). For the aggressive step, we use a multiplier $[10, 1]$ with a polynomial decay rate with power= 0.5 for the first $10K$ iterations. In all cases, the learning rate of the classifiers is set 10 times larger than the one of the feature extractor (0.01) whose value is adjusted according to Ganin et al. (2016), which is standard practice. We will release source code.

**Comparing $f$-divergences.** We first compare the performance of $f$-divergences on *Office-31*. Specifically, we evaluate the performance of the model on the six combinations of transfer tasks with different divergences. The optimizer is SGD with Nesterov Momentum. All hyperparameters are kept constant for all divergences. As shown in Figure 4, the *Pearson $\chi^2$ achieves the best overall result among all the transfer tasks on this benchmark*. This divergence was never used before to learn invariant representations in the context of DA. Interestingly, a similar trend was observed for GANs in Nowozin et al. (2016). This observation is also reminiscent of histogram-based (visual) bag of words representations

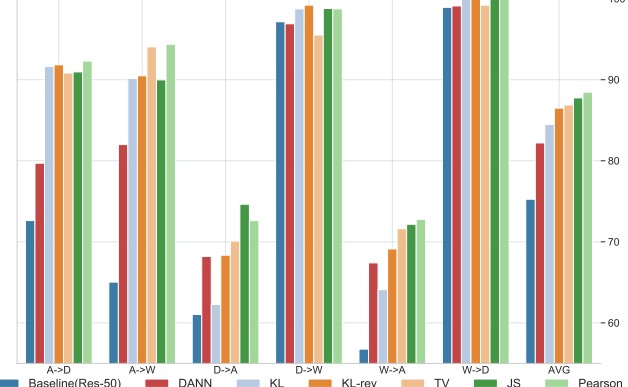

Figure 4: Transfer performance of a model trained using $f$-DAL for different choices of divergences and different transfer tasks on the Office-31 benchmark. Baseline is ResNet-50 w/o $f$-DAL. We additionally show the performance of DANN (Table 2). When compared with $f$-DAL (JS), we see a significant performance boost. This is in line with our theory which suggests the use of a per-category domain classifier vs a discriminator.

that were shown to work better with $\chi^2$ distances than with $\ell_2$ and $\ell_1$ distances for image and text classification tasks (i.e. (Li et al., 2013) and references therein). The KL divergence performs well in some transfer tasks but significantly worse than the rest in others (i.e $\mathbf{D} \to \mathbf{A}$). The reason might be that, unlike the JS, TV and Pearson $\chi^2$ divergences which are lower and upper bounded by finite values, the KL divergence can grow exponentially and tend to $+\infty$ even when the densities $p_s$ and $p_t$ are nonzero (Nielsen & Nock, 2013). The lack of upper bound of the KL divergence might lead to numerical instability of the optimizer and explain inconsistency of performance.

**What do we get by the "*extra*" gradient?** We now compare the use of the AExG vs GDA method with Nesterov Momentum. The main idea is to evaluate whether the characterization of the optimal solution of $f$-DAL as a Stackelberg Equilibrium leads to practical gains by exploiting more suitable

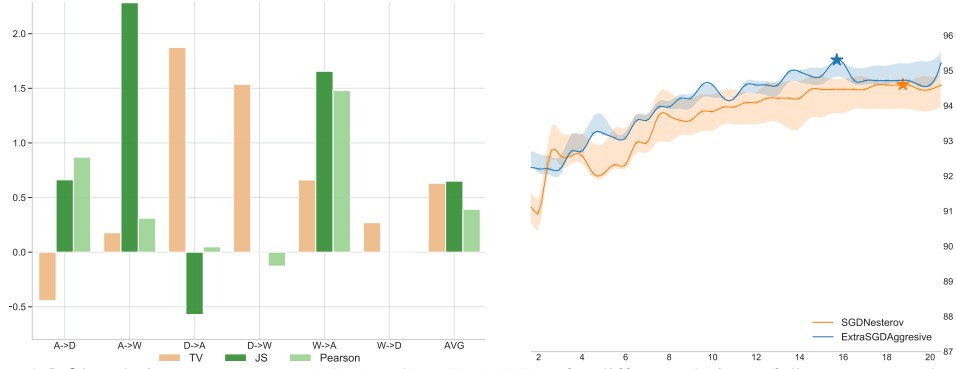

Figure 5: **left)** Relative Improvement (%) AExG vs GDA(SGD) for different choices of divergences and transfer tasks on the Office-31 benchmark. Overall, we observe gains in performance among all divergences. **right)** Transfer curves for Pearson $\chi^2$ on the task A→W on the Office-31 benchmark (# Iter vs Acc). We can see AExG converges faster and also obtains slightly better results. This is inline with the insights obtained from the theoretical results presented in Sec 4.1 and Appendix 5.

| Method | Ar→Cl | Ar→Pr | Ar→Rw | Cl→Ar | Cl→Pr | Cl→Rw | Pr→Ar | Pr→Cl | Pr→Rw | Rw→Ar | Rw→Cl | Rw→Pr | Avg |
|---|---|---|---|---|---|---|---|---|---|---|---|---|---|
| ResNet-50 (He et al., 2016) | 34.9 | 50.0 | 58.0 | 37.4 | 41.9 | 46.2 | 38.5 | 31.2 | 60.4 | 53.9 | 41.2 | 59.9 | 46.1 |
| DANN (Ganin et al., 2016) | 45.6 | 59.3 | 70.1 | 47.0 | 58.5 | 60.9 | 46.1 | 43.7 | 68.5 | 63.2 | 51.8 | 76.8 | 57.6 |
| JAN (Long et al., 2017) | 45.9 | 61.2 | 68.9 | 50.4 | 59.7 | 61.0 | 45.8 | 43.4 | 70.3 | 63.9 | 52.4 | 76.8 | 58.3 |
| CDAN (Long et al., 2018) | 50.7 | 70.6 | 76.0 | 57.6 | 70.0 | 70.0 | 57.4 | 50.9 | 77.3 | 70.9 | 56.7 | 81.6 | 65.8 |
| MDD (Zhang et al., 2019) | 54.9 | **73.7** | 77.8 | 60.0 | 71.4 | 71.8 | 61.2 | 53.6 | 78.1 | 72.5 | 60.2 | 82.3 | 68.1 |
| Ours (Pearson $\chi^2$, SGD) | 54.7 | 69.4 | 77.8 | **61.0** | 72.6 | 72.2 | 60.8 | 53.4 | 80.0 | 73.3 | 60.6 | **83.8** | 68.3 |
| Ours (Pearson $\chi^2$, AExG) | **55.2** | 70.2 | **78.6** | 60.9 | **73.2** | **72.8** | **61.3** | **53.6** | **80.5** | **73.7** | **61.0** | 83.6 | **68.7** |

Table 3: Accuracy (%) on Office-Home for unsupervised DA. Impressively, our approach achieves SoTA without additional techniques (i.e. CDAN), or additional hyperparameters (i.e. MDD).

optimizers (Section 4.1). In both cases, we use a momentum coefficient (0.9). The experiments are performed on Office-31 and hyperparameters are kept constant for all divergences. In Figure 5, we observe that using AExG significantly improves the performance in some transfer tasks for some divergences (i.e JS in A → W). Overall, we also observe gains in performance among all divergences. Figure 5 illustrates the transfer curves of the AExG vs GDA with Nesterov Momentum for the task A→ W. For this divergence and pair of datasets, AExG converges faster and also obtains slightly better accuracy. This is in line with the insights obtained from the theoretical results presented in Section 4.1 and Appendix 5. If computation is not an issue we encourage the use of AExG. That said, $f$-DAL achieves comparable performance with GDA in terms of accuracy (see Tables 2 and 3).

**A look to state-of-the-art arena.** We also compare our best method (i.e Pearson $\chi^2$ + AExG and Pearson $\chi^2$ + GDA) with current SoTA unsupervised DA methods. We use this in a *leader-board like fashion*. For fair comparison, we use the same network architecture (i.e Resnet-50), training strategy and set of hyperparameters from Long et al. (2018); Zhang et al. (2019) from where we took the baselines results. Our approach achieves SoTA results on Office-31 and Office-Home Datasets.

**Remark on SoTA Comparison:** We compare with SoTAs that rely on adversarial training since this is the focus of our work. Therefore, methods such as Kang et al. (2019) are not included as they rely on additional techniques that neither our method nor the proposed baselines use and could be added to improve the performance further. Our goal is to propose a unifying framework that connects the theory used to explain DANN (Ganin et al., 2016) (and similar SoTA algorithms), and the algorithms themselves. The new theory results in a new adversarial framework (Sec 4), which impressively outperforms previous SoTA (Tables 2 and 3). This follows from connecting theory and algorithms and a proper interpretation of the former. Our results can be further improved with additional tuning or techniques (i.e CDAN) since most of SoTAs either follow from Ganin et al. (2016) or are part of our framework (i.e MDD= $\gamma$-JS). This problem is deferred to future work.

## 6 CONCLUSIONS

We have provided a novel perspective on the domain-adversarial problem by deriving new theory and learning algorithms that support the complete family of $f$-divergences, and that are practical for modern neural networks. We further recognize the learning objective of our framework as a Stackelberg game, borrowing the latest optimizers from the game-optimization literature, achieving additional performance boosts. We show through large-scale experiments that any $f$-divergence can be used to minimize the discrepancy between source and target domains in a representation space. We also show that some divergences, not considered previously in domain-adversarial learning, achieve SoTA results in practice, reducing the need for additional techniques and hyperparameter tuning as required by previous methods.

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

# A    RELATED WORK

## A.1    DOMAIN ADAPTATION

We briefly but clearly positioned our approach w.r.t. related work mentioned in the paper. We refer the reader to Redko et al. (2019) and Wang & Deng (2018) for a comprehensive survey.

## A.2    DIVERGENCES BETWEEN PROBABILITY MEASURES

As explained above, the difference term between source and target domains is important in bounding the target loss. We now provide more details about the $\mathcal{H}\Delta\mathcal{H}$-divergence and $f$-divergences that are used to compare both domains.

$\mathcal{H}\Delta\mathcal{H}$**-divergence**    The $\mathcal{H}$-divergence is a restriction of total variation. For binary classification, define $I(h) := \{\mathbf{x} \in \mathcal{X} : h(\mathbf{x}) = 1\}$, then the $\mathcal{H}$-divergence between two measures $\mu$ and $\nu$ given the hypothesis class $\mathcal{H}$ is (Ben-David et al., 2010a):

$$d_{\mathcal{H}}(\mu, \nu) = 2 \sup_{h \in \mathcal{H}} |\mu(I(h)) - \nu(I(h))|. \tag{A.1}$$

Define $\mathcal{H}\Delta\mathcal{H} := \{h \oplus h' : h, h' \in \mathcal{H}\}$ ($\oplus$: XOR), then $d_{\mathcal{H}\Delta\mathcal{H}}(\mu, \nu)$ can be used to bound the difference between the source and target errors. $\mathcal{H}\Delta\mathcal{H}$ divergence has been extended to general loss functions (Mansour et al., 2009) and marginal disparity discrepancy (Zhang et al., 2019).

$f$**-divergence**    Given two measures $\mu$ and $\nu$ with $\mu \ll \nu$ ($\mu$ absolute continuous wrt $\nu$), the $f$-divergence $D_\phi(\mu||\nu)$ is defined as (Csiszár, 1967; Ali & Silvey, 1966):

$$D_\phi(\mu \parallel \nu) = \int \phi\left(\frac{d\mu}{d\nu}\right) d\nu, \tag{A.2}$$

where $d\mu/d\nu$ is known as the Radon–Nikodym derivative (e.g. Billingsley, 2008). Assume $\phi$ is convex and lower semi-continuous, then from the Fenchel–Moreau theorem, $\phi^{**} = \phi$, with $\phi^*$ known as the Fenchel conjugate of $\phi$:

$$\phi^*(\mathbf{y}) = \sup_{\mathbf{x} \in \text{dom } \phi} \langle \mathbf{x}, \mathbf{y} \rangle - \phi(\mathbf{x}), \tag{A.3}$$

which is convex since it is a supremum of an affine function. In order for $\mathbf{x}$ to take the supremum, it is necessary and sufficient that $\mathbf{y} \in \partial\phi(\mathbf{x})$ using the stationarity condition. Therefore, with equation A.2 and equation A.3, $D_\phi(\mu \parallel \nu)$ can be written as:

$$D_\phi(\mu \parallel \nu) = \sup_{T \in \mathcal{T}} \mathbb{E}_{X \sim \mu}[T(X)] - E_{Z \sim \nu}[\phi^*(T(Z))], \tag{A.4}$$

where $\mathcal{T} = \{T : T$ is a measurable function and $T : \mathcal{X} \to \text{dom } \phi^*\}$. In practice we restrict $\mathcal{T}$ to a subset as in Definition 2. For different choices of $\phi$ see Table 4.

Nguyen et al. (2010) derive a general variational method to estimate $f$-divergences given only samples. Nowozin et al. (2016) extend their method from merely estimating a divergence for a fixed model to estimating model parameters. While our method builds on this variational formulation, we use it in the context of domain adaptation.

# B    PROOFS

In this section, we provide the proofs for the different theorems and lemmas:

**Theorem 1.** *If $\ell(x, y) = |h(x) - y|$ and $\mathcal{H}$ is a class of functions, then for any $h \in \mathcal{H}$ we have:*

$$R_T^\ell(h) \leq R_S^\ell(h) + D_{\text{TV}}(P_s \| P_t) + \min\{\mathbb{E}_{x \sim P_s}[|f_t(x) - f_s(x)|], \mathbb{E}_{x \sim P_t}[|f_t(x) - f_s(x)|]\}. \tag{3.1}$$

*Proof.* Rewriting the target loss we have:

$$R_T^\ell(h) = R_T^\ell(h) - R_S^\ell(h, f_t) + R_S^\ell(h, f_t) - R_S^\ell(h) + R_S^\ell(h),$$
$$\leq R_S^\ell(h) + |R_S^\ell(h) - R_S^\ell(h, f_t)| + |R_T^\ell(h) - R_S^\ell(h, f_t)|$$

| Divergence | $\phi(x)$ | $\phi^*(t)$ | $\phi'(1)$ | $g(x)$ |
|---|---|---|---|---|
| MDD | $x \log \frac{\gamma x}{1+\gamma x} + \frac{1}{\gamma} \log \frac{1}{1+\gamma x}$ | $-\log(1-e^t)/\gamma$ | $\log \frac{\gamma}{1+\gamma}$ | $\log x$ |
| Kullback-Leibler (KL) | $x \log x$ | $\exp(t-1)$ | $1$ | $x$ |
| Reverse KL (KL-rev) | $- \log x$ | $-1 - \log(-t)$ | $-1$ | $-\exp x$ |
| Jensen-Shannon (JS) | $-(x+1) \log \frac{1+x}{2} + x \log x$ | $-\log(2 - e^t)$ | $0$ | $\log \frac{2}{1+\exp(-x)}$ |
| Pearson $\chi^2$ | $(x-1)^2$ | $t^2/4 + t$ | $0$ | $x$ |
| Squared Hellinger (SH) | $(\sqrt{x} - 1)^2$ | $\frac{t}{1-t}$ | $0$ | $1 - \exp x$ |
| $\gamma$-weighted Pearson $\chi^2$ | $(\gamma x - 1)^2/\gamma$ | $(t^2/4 + t)/\gamma$ | $0$ | $x$ |
| Neynman $\chi^2$ | $\frac{(1-x)^2}{x}$ | $2 - 2\sqrt{1-t}$ | $0$ | $1 - \exp x$ |
| $\gamma$-weighted total variation | $\frac{1}{2\gamma} \lvert \gamma x - 1 \rvert$ | $(t/\gamma)\mathbf{1}_{-1/2 \leq t \leq 1/2}$ | $[-1/2, 1/2]$ | $\frac{1}{2} \tanh x$ |
| Total Variation (TV) | $\frac{1}{2} \lvert x - 1 \rvert$ | $\mathbf{1}_{-1/2 \leq t \leq 1/2}$ | $[-1/2, 1/2]$ | $\frac{1}{2} \tanh x$ |

Table 4: Popular $f$-divergences, their conjugate functions and choices of $g$. We take $\hat{l}(a,b) = g(b_{\mathrm{argmax}\, a})$.

where:

$$
\begin{aligned}
|R_S^\ell(h) - R_S^\ell(h, f_t)| &= |R_S^\ell(h, f_s) - R_S^\ell(h, f_t)| \\
&= |\mathbb{E}_{x \sim P_s}[|h(x) - f_t(x)| - |h(x) - f_s(x)|]| \\
&\leq \mathbb{E}_{x \sim P_s}[|f_t(x) - f_s(x)|]
\end{aligned}
$$

and:

$$
\begin{aligned}
|R_T^\ell(h) - R_S^\ell(h, f_t)| = |R_T^\ell(h, f_t) - R_S^\ell(h, f_t)| &\leq \int |p_{\mathrm{t}}(x) - p_{\mathrm{s}}(x)| \cdot |h(x) - f_t(x)| dx \\
&\leq \int |(\frac{p_{\mathrm{t}}(x)}{p_{\mathrm{s}}(x)} - 1)p_{\mathrm{s}}(x)| dx = D_\phi(P_s || P_{\mathrm{t}})
\end{aligned}
$$

with $\phi(x) = |x - 1|$ which represents the total divergence. $\qquad\square$

**Lemma 1 (lower bound).** *For any two functions h,h' in $\mathcal{H}$, we have:*

$$
|R_S^\ell(h, h') - R_T^{\phi^* \circ \ell}(h, h')| \leq D_{h, \mathcal{H}}^\phi(P_s || P_t) \leq D_{\mathcal{H}}^\phi(P_s || P_t) \leq D_\phi(P_s || P_t). \tag{3.4}
$$

*Proof.*

$$
\begin{aligned}
\mathrm{D}_{\mathcal{H}}^\phi(P_{\mathrm{s}} || P_{\mathrm{t}}) &= \sup_{h \in \mathcal{H}} \mathrm{D}_{h, \mathcal{H}}^\phi(P_{\mathrm{s}} || P_{\mathrm{t}}) \geq \mathrm{D}_{h, \mathcal{H}}^\phi(P_{\mathrm{s}} || P_{\mathrm{t}}) && \text{(B.1)} \\
&= \sup_{h' \in \mathcal{H}} |E_{x \sim P_{\mathrm{s}}}[\ell(h(x), h'(x))] - \mathbb{E}_{x \sim P_{\mathrm{t}}}[\phi^*(\ell(h(x), h'(x)))]| && \text{(B.2)} \\
&\geq |E_{x \sim P_{\mathrm{s}}}[\ell(h(x), h'(x))] - \mathbb{E}_{x \sim P_{\mathrm{t}}}[\phi^*(\ell(h(x), h'(x)))]| && \text{(B.3)} \\
&= |R_S^\ell(h, h') - R_T^{\phi^* \circ \ell}(h, h')|. && \text{(B.4)}
\end{aligned}
$$

For the rightmost inequality in equation 3.4, it is well-known that $f$-divergence $D_\phi$ is nonnegative (e.g. Sason & Verdú, 2016), and thus

$$
D_\phi(P_{\mathrm{s}} || P_{\mathrm{t}}) = \sup_{T \in \mathcal{T}} |\mathbb{E}_{x \sim P_{\mathrm{s}}} T(x) - \mathbb{E}_{x \sim P_{\mathrm{t}}} \phi^*(T(x))|. \tag{B.5}
$$

Restricting $\mathcal{T}$ to $\hat{\mathcal{T}}$ as in Definition 2 we obtain $D_\phi(P_{\mathrm{s}} || P_{\mathrm{t}}) \geq \mathrm{D}_{\mathcal{H}}^\phi(P_{\mathrm{s}} || P_{\mathrm{t}})$. $\qquad\square$

**Lemma 2.** *Suppose $\ell : \mathcal{Y} \times \mathcal{Y} \to [0, 1]$, $\phi^*$ L-Lipschitz, and $[0, 1] \subset \mathrm{dom}\, \phi^*$. Let S and T be two empirical distributions corresponding to datasets containing $n$ datapoints sampled i.i.d. from $P_s$ and $P_t$, respectively. Let us note $\mathfrak{R}$ the Rademacher complexity of a given class of functions, and $\ell \circ \mathcal{H} := \{x \mapsto \ell(h(x), h'(x)) : h, h' \in \mathcal{H}\}$. $\forall \delta \in (0, 1)$, we have with probability of at least $1 - \delta$:*

$$
|D_{h, \mathcal{H}}^\phi(P_s || P_t) - D_{h, \mathcal{H}}^\phi(S || T)| \leq 2\mathfrak{R}_{P_s}(\ell \circ \mathcal{H}) + 2L\mathfrak{R}_{P_t}(\ell \circ \mathcal{H}) + 2\sqrt{(-\log \delta)/(2n)}. \tag{3.5}
$$

*Proof.* For reference, we refer the reader to Chapter 3 of Mohri et al. (2018). Using the notations of $R$ and $\hat{R}$ that represent the true and empirical risks, we have:

$$
\begin{aligned}
\mathrm{D}^{\phi}_{h,\mathcal{H}}(P_s||P_t) - \mathrm{D}^{\phi}_{h,\mathcal{H}}(\mathrm{S}||\mathrm{T}) &= \sup_{h' \in \mathcal{H}} \{|R^{\ell}_S(h,h') - R^{\phi^* \circ \ell}_T(h,h')|\} \\
&\quad - \sup_{h' \in \mathcal{H}} \{|\hat{R}^{\ell}_S(h,h') - \hat{R}^{\phi^* \circ \ell}_T(h,h')|\} \\
&\leq \sup_{h' \in \mathcal{H}} ||R^{\ell}_S(h,h') - R^{\phi^* \circ \ell}_T(h,h')| - |\hat{R}^{\ell}_S(h,h') - \hat{R}^{\phi^* \circ \ell}_T(h,h')|| \\
&\leq \sup_{h' \in \mathcal{H}} |R^{\ell}_S(h,h') - R^{\phi^* \circ \ell}_T(h,h') - \hat{R}^{\ell}_S(h,h') + \hat{R}^{\phi^* \circ \ell}_T(h,h')| \\
&= \sup_{h' \in \mathcal{H}} |R^{\ell}_S(h,h') - \hat{R}^{\ell}_S(h,h')| + |R^{\phi^* \circ \ell}_T(h,h') - \hat{R}^{\phi^* \circ \ell}_T(h,h')| \\
&\leq 2\mathfrak{R}_{P_s}(\ell \circ \mathcal{H}) + \sqrt{\frac{\log \frac{1}{\delta}}{2n}} + 2\mathfrak{R}_{P_t}(\phi^* \circ \ell \circ \mathcal{H}) + \sqrt{\frac{\log \frac{1}{\delta}}{2n}}
\end{aligned}
\tag{B.6}
$$

where: $|R^{\ell}_S(h,h') - \hat{R}^{\ell}_S(h,h')| \leq 2\mathfrak{R}_{P_s}(\ell \circ \mathcal{H}) + \sqrt{\frac{\log \frac{1}{\delta}}{2n}}$ (Theorem 3.3 of Mohri et al. (2018)). Similarly, by Talagrand's lemma (Lemma 5.7 and Definition 3.2 of Mohri et al. (2018)) we have: $\mathfrak{R}_{P_t}(\phi^* \circ \ell \circ \mathcal{H}) \leq \mathrm{L}\mathfrak{R}_{P_t}(\ell \circ \mathcal{H})$, with $\phi^* \circ \ell \circ \mathcal{H} := \{x \mapsto \phi(\ell(h(x), h'(x))) : h, h' \in \mathcal{H}\}$. $\qquad \square$

**Theorem 2 (generalization bound).** *Suppose* $\ell : \mathcal{Y} \times \mathcal{Y} \to [0,1] \subset \mathrm{dom}\, \phi^*$ *and that* $\ell(a,b) \leq \ell(a,c) + \ell(c,b)$ *for any* $a,b,c \in \mathcal{Y}$. *Denote* $\lambda^* := R^{\ell}_S(h^*) + R^{\ell}_T(h^*)$, *and let* $h^*$ *be the ideal joint hypothesis. We have:*

$$
R^{\ell}_T(h) \leq R^{\ell}_S(h) + D^{\phi}_{h,\mathcal{H}}(P_s||P_t) + \lambda^*.
\tag{3.6}
$$

*Proof.* We first introduce the following lemma for our proof:

**Lemma 3.** *For any function* $\phi$ *that satisfies* $\phi(1) = 0$ *we have* $\phi^*(t) \geq t$ *where* $\phi^*$ *is the Fenchel conjugate of* $\phi$.

*Proof.* From the definition of Fenchel conjugate, $\phi^*(t) = \sup_{x \in \mathrm{dom}\, \phi}(xt - \phi(x)) \geq t - \phi(1) = t$. $\qquad \square$

With the triangle inequality of $\ell$, we can write:

$$
\begin{aligned}
R^{\ell}_T(h, f_t) &\leq R^{\ell}_T(h, h^*) + R^{\ell}_T(h^*, f_t) && \text{(B.7)} \\
&= R^{\ell}_T(h, h^*) + R^{\ell}_T(h^*, f_t) - R^{\ell}_S(h, h^*) + R^{\ell}_S(h, h^*) && \text{(B.8)} \\
&\leq R^{\phi^* \circ \ell}_T(h, h^*) - R^{\ell}_S(h, h^*) + R^{\ell}_S(h, h^*) + R^{\ell}_T(h^*, f_t) && \text{(Lemma 3)} && \text{(B.9)} \\
&\leq |R^{\phi^* \circ \ell}_T(h, h^*) - R^{\ell}_S(h, h^*)| + R^{\ell}_S(h, h^*) + R^{\ell}_T(h^*, f_t) && \text{(B.10)} \\
&\leq D^{\phi}_{h,\mathcal{H}}(P_s||P_t) + R^{\ell}_S(h, h^*) + R^{\ell}_T(h^*, f_t) && \text{(Lemma 1)} && \text{(B.11)} \\
&\leq D^{\phi}_{h,\mathcal{H}}(P_s||P_t) + R^{\ell}_S(h, f_s) + \underbrace{R^{\ell}_S(h^*, f_s) + R^{\ell}_T(h^*, f_t)}_{\lambda^*}. && \text{(B.12)}
\end{aligned}
$$

$$\square$$

**Theorem 3 (generalization bound with Rademacher complexity).** *Let* $\ell : \mathcal{Y} \times \mathcal{Y} \to [0,1]$ *and* $\phi^*$ *be L-Lipschitz. Let S and T be two empirical distributions (i.e. datasets containing* $n$ *data points sampled i.i.d. from* $P_s$ *and* $P_t$, *respectively). Denote* $\hat{\lambda}^*_{\phi} := \hat{R}^{\ell}_S(h^*) + \hat{R}^{\ell}_T(h^*)$. $\forall \delta \in (0,1)$, *we have with probability of at least* $1 - \delta$:

$$
\begin{aligned}
R^{\ell}_T(h) &\leq \hat{R}^{\ell}_S(h) + D^{\phi}_{h,\mathcal{H}}(S||T) + \hat{\lambda}^*_{\phi} \\
&\quad + 6\mathfrak{R}_S(\ell \circ \mathcal{H}) + 2(1 + L)\mathfrak{R}_T(\ell \circ \mathcal{H}) + 5\sqrt{(-\log \delta)/(2n)}.
\end{aligned}
\tag{3.7}
$$

*Proof.* We show in the following that:

$$R_T^\ell(h) \leq \hat{R}_S^\ell(h) + \mathrm{D}_{h,\mathcal{H}}^\phi(\mathrm{S}||\mathrm{T}) + \hat{\lambda}_\phi^* \tag{B.13}$$

$$+ 6\Re_S(\ell \circ \mathcal{H}) + 2(1+L)\Re_T(\ell \circ \mathcal{H}) + 5\sqrt{(-\log \delta)/(2n)}. \tag{B.14}$$

This follows from Theorem 2 where: $R_T^\ell(h) \leq R_S^\ell(h) + \mathrm{D}_{h,\mathcal{H}}^\phi(P_s||P_t) + R_S^\ell(h^*) + R_T^\ell(h^*)$. We also have: $|R_D^\ell(h) - \hat{R}_D^\ell(h)| \leq 2\Re_D(\ell \circ \mathcal{H}) + \sqrt{\frac{\log \frac{1}{\delta}}{2n}}$ (Theorem of 3.3 Mohri et al. (2018)). From Lemma 2, $\mathrm{D}_{h,\mathcal{H}}^\phi(P_s||P_t) \leq 2\Re_{P_s}(\ell \circ \mathcal{H}) + 2\mathrm{L}\Re_{P_t}(\ell \circ \mathcal{H}) + 2\sqrt{\frac{\log \frac{1}{\delta}}{2n}}$. Plugging in and rearranging gives the desired results. □

**Proposition 1.** *Suppose $d_{s,t}$ takes the form shown in equation 4.2 with $\hat{\ell}(\hat{h}'(z), \hat{h}(z)) \rightarrow \mathrm{dom}\,\phi^*$ and that for any $\hat{h} \in \hat{\mathcal{H}}$, there exists $\hat{h}' \in \hat{\mathcal{H}}$ s.t. $\hat{\ell}(\hat{h}'(z), \hat{h}(z)) = \phi'(\frac{p_s^z(z)}{p_t^z(z)})$ for any $z \in \mathrm{supp}(p_t^z(z))$, with $\phi'$ the derivative of $\phi$. The optimal $d_{s,t}$ is $D_\phi(P_s^z||P_t^z)$ (i.e $\max_{\hat{h}' \in \hat{\mathcal{H}}} d_{s,t} = D_\phi(P_s^z||P_t^z)$).*

*Proof.* We first rewrite from the definition of $d_{s,t}$ in equation 4.2:

$$d_{s,t} = \mathbb{E}_{z \sim p_s^z}[\hat{\ell}(\hat{h}'(z), \hat{h}(z))] - \mathbb{E}_{z \sim p_t^z}[(\phi^* \circ \hat{\ell})(\hat{h}'(z), \hat{h}(z))] \tag{B.15}$$

$$= \int [p_s^z(z)\hat{\ell}(\hat{h}'(z), \hat{h}(z)) - p_t^z(z)(\phi^* \circ \hat{\ell})(\hat{h}'(z), \hat{h}(z))]dz \tag{B.16}$$

$$= \int p_t^z(z) \left[ \frac{p_s^z(z)}{p_t^z(z)}\hat{\ell}(\hat{h}'(z), \hat{h}(z)) - (\phi^* \circ \hat{\ell})(\hat{h}'(z), \hat{h}(z)) \right] dz. \tag{B.17}$$

Maximizing w.r.t $h'$ and assuming $\hat{\mathcal{H}}$ is unconstrained we have: $\frac{p_s^z(z)}{p_t^z(z)} \in (\partial\phi^*)(\hat{\ell}(\hat{h}'(z), \hat{h}(z))$ for any $z \in \mathrm{supp}(p_t^z)$. From the definition of Fenchel conjugate we have:

$$x \in \partial\phi^*(t) \iff \phi(x) + \phi^*(t) = xt \iff \phi'(x) = t.$$

Plugging $x = p_s^z(z)/p_t^z(z)$ and $t = \ell(\hat{h}'(z), \hat{h}(z))$ we obtain $\ell(\hat{h}'(z), \hat{h}(z)) = \phi'(p_s^z(z)/p_t^z(z))$. Hence, from the definition of $f$-divergences (Definition 1) and its variational characterization (eq. 2.2), we write:

$$\max_{\hat{h}' \in \hat{\mathcal{H}}} d_{s,t} = D_\phi(P_s^z||P_t^z). \tag{B.18}$$

□

## C    THE EXISTENCE OF THE STACKELBERG/NASH EQUILIBRIUM IN $f$-DAL

In this appendix, we formally define the Nash/Stackelberg equilibrium and show that they exist in our $f$-DAL framework under mild assumptions.

**Definition 4** (**Stackelberg equilibrium**). *A Stackelberg equilibrium $(\omega_1^*, \omega_2^*) \in \Omega_1 \times \Omega_2$ of the min-max game satisfies $\forall(\omega_1, \omega_2) \in \Omega_1 \times \Omega_2$, $V(\omega_1^*, \omega_2) \leq V(\omega_1^*, \omega_2^*) \leq \max_{\omega_2 \in \Omega_2} V(\omega_1, \omega_2)$.*

**Definition 5** (**Nash equilibrium**). *A Nash equilibrium $(\omega_1^*, \omega_2^*) \in \Omega_1 \times \Omega_2$ of the min-max game equation 4.4 is defined such that $\forall(\omega_1, \omega_2) \in \Omega_1 \times \Omega_2$, $V(\omega_1^*, \omega_2) \leq V(\omega_1^*, \omega_2^*) \leq V(\omega_1, \omega_2^*)$.*

**Theorem 5** (**Stackelberg equilibrium**). *Suppose $d_{s,t}$ takes the form shown in equation 4.2, and assume that (a) There is an optimal feature extractor $g^* \in \mathcal{G}$ that maps both the source and the target distribution to the same distribution, i.e. $g^*\#p_s = g^*\#p_t$. (b) There is an optimal classifier s.t. $\hat{h}^* \circ g^* = f_s$ is the ground truth, and $\hat{h} \circ g = f_s$ for any $(\hat{h}, g)$ in a neighborhood of $(\hat{h}^*, g^*)$. (c) For any $g \in \mathcal{G}$ and $\hat{h} \in \mathcal{H}$, there exists $\hat{h}'$ s.t. for any $z \in \mathrm{supp}(g\#p_s)$, one has $\hat{\ell}(\hat{h}'(z), \hat{h}(z)) = \phi'((g\#p_s)(z)/(g\#p_t)(z))$. Then the objective of $f$-adversarial learning has a Stackelberg equilibrium at $(\hat{h}^*, g^*, \hat{h}'^*)$, where $\forall z \in \mathrm{supp}(g^*\#p_s)$, $\hat{\ell}(\hat{h}^*(z), \hat{h}'^*(z)) = \phi'(1)$.*

*Proof.* At $g^*$ and $\hat{h}^*$, we have

$$d_{s,t}(\hat{h}^*, g^*, \hat{h}') = \mathbb{E}_{z \sim g^*\#p_s}\hat{\ell}(\hat{h}^*(z), \hat{h}'(z)) - \phi^*(\hat{\ell}(\hat{h}^*(z), \hat{h}'(z))). \tag{C.1}$$

Maximizing over $\hat{\ell}(\hat{h}^*(z), \hat{h}'(z))$ yields:

$$\hat{\ell}(\hat{h}^*(z), \hat{h}'^*(z)) = \phi'(1), \forall z \in \text{supp}(g^* \# p_s). \tag{C.2}$$

In other words, $d_{s,t}(\hat{h}^*, g^*, \hat{h}') \leq d_{s,t}(\hat{h}^*, g^*, \hat{h}'^*)$ for any $\hat{h}' \in \hat{\mathcal{H}}$. Now let us prove that $\max_{\hat{h}' \in \hat{\mathcal{H}}} d_{s,t}(\hat{h}, g, \hat{h}') \geq d_{s,t}(\hat{h}^*, g^*, \hat{h}'^*) = \phi(1) = 0$. This is because from our assumptions and Lemma 1, $\max_{\hat{h}' \in \hat{\mathcal{H}}} d_{s,t}(\hat{h}, g, \hat{h}') = D_\phi(g \# p_s \| g \# p_t) \geq 0$.

So far, we have shown that

$$d_{s,t}(\hat{h}^*, g^*, \hat{h}') \leq d_{s,t}(\hat{h}^*, g^*, \hat{h}'^*) \leq \max_{\hat{h}' \in \hat{\mathcal{H}}} d_{s,t}(\hat{h}, g, \hat{h}') \tag{C.3}$$

for any $\hat{h}, \hat{h}' \in \mathcal{H}$ and $g \in \mathcal{G}$. Also, $(\hat{h}^*, g^*)$ is an optimal pair for the source loss, namely:

$$R_s(\hat{h}^*, g^*) \leq R_s(\hat{h}, g), \tag{C.4}$$

for any $\hat{h} \in \mathcal{H}$ and $g \in \mathcal{G}$. Combining equation C.3 and equation C.8, we claim that $(\hat{h}^*, g^*, \hat{h}'^*)$ is a Stackelberg equilibrium. $\qquad\square$

**Theorem 6** (**Nash equilibrium**). *With the same assumptions as in Theorem 5 and assume also that $\ell$ only depends on the second argument, i.e., $\ell(\hat{h}, \hat{h}') = \ell(\hat{h}')$. Then the objective of $f$-adversarial learning has a Nash equilibrium at $(\hat{h}^*, g^*, \hat{h}'^*)$ where*

$$\forall z \in \bigcup_{g \in \mathcal{G}} \text{supp}(g \# p_s), \ \hat{\ell}(\hat{h}'^*(z)) = \phi'(1). \tag{C.5}$$

*Proof.* The proof is in parallel to the proof of Theorem 5 except that at $\hat{h}'^*$,

$$
\begin{aligned}
d_{s,t}(g, \hat{h}'^*) &= \mathbb{E}_{z \sim p_s^z} \hat{\ell}(\hat{h}'^*(z)) - \mathbb{E}_{z \sim p_t^z} \phi^* \circ \hat{\ell}(\hat{h}'^*(z)) \\
&= \phi'(1) - \phi^*(\phi'(1))
\end{aligned}
\tag{C.6}
$$

is a constant in terms of $g$, and thus we have:

$$d_{s,t}(g, \hat{h}'^*) \geq d_{s,t}(g^*, \hat{h}'^*) \geq d_{s,t}(g^*, \hat{h}'), \tag{C.7}$$

for any $g \in \mathcal{G}$ and $\hat{h}'^* \in \hat{\mathcal{H}}$. Combining with equation C.8 we conclude that $(\hat{h}^*, g^*, \hat{h}'^*)$ is a Nash equilibrium of $d_{s,t}$. Also, $(\hat{h}^*, g^*)$ is an optimal pair for the source loss, namely:

$$R_s(\hat{h}^*, g^*) \leq R_s(\hat{h}, g), \tag{C.8}$$

for any $\hat{h} \in \mathcal{H}$ and $g \in \mathcal{G}$. Combining equation C.7, we claim that $(\hat{h}^*, g^*, \hat{h}'^*)$ is a Nash equilibrium of the objective in $f$-DAL. $\qquad\square$

## D    CONNECTION TO PREVIOUS FRAMEWORKS

In this appendix we show that $f$-DAL encompasses previous frameworks on domain adaptation, including $\mathcal{H}\Delta\mathcal{H}$-divergence, DANN (Ganin et al., 2016) and MDD (Zhang et al., 2019).

### D.1    $\mathcal{H}\Delta\mathcal{H}$-DIVERGENCE

We now show that Theorem 2 generalizes the bound proposed in Ben-David et al. (2010a). Let the pair $\{\phi(x), \phi^*(t)\} = \{\frac{1}{2}|x-1|, t\}$ for $t \in [0,1]$, such that $D_{h,\mathcal{H}}^\phi = D_{h,\mathcal{H}}^{\text{TV}}$ and $\sup_{h \in \mathcal{H}} D_{h,\mathcal{H}}^{\text{TV}} = D_{\mathcal{H}}^{\text{TV}} = \frac{1}{2} d_{\mathcal{H}\Delta\mathcal{H}}$, with $d_{\mathcal{H}\Delta\mathcal{H}}$ defined in Ben-David et al. (2010a) (see also equation A.1). Theorem 2 gives us that $R_T^\ell(h) \leq R_S^\ell(h) + \frac{1}{2} d_{\mathcal{H}\Delta\mathcal{H}} + \lambda^*$, recovering Theorem 2 of Ben-David et al. (2010a).

### D.2 DANN FORMULATION AND JS DIVERGENCE

The DANN formulation by Ganin & Lempitsky (2015) can also be incorporated in our framework if one takes $\hat{\ell}(a, b) = \log b$ and $\phi^*(t) = -\log(1 - e^t)$. Effectively, this formulation ignores the contribution of the source classifier and experimentally we saw it had inferior performance compared to using $\hat{\ell}(a, b) = g(b_{\text{argmax } a})$.

The original idea of domain adversarial training was introduced by Ganin et al. (2016) where the authors defined the following surrogate function to measure the discrepancy between the two domains:

$$d_{s,t} := \mathbb{E}_{x_s \sim p_s}[\log \hat{h}'(g(x_s))] + \mathbb{E}_{x_t \sim p_t}[\log(1 - \hat{h}'(g(x_t)))]. \tag{D.1}$$

In this context, $\hat{h}'$ was defined to be a domain classifier, that is $\hat{h}' : \mathcal{Z} \to \{0, 1\}$ with $0$ and $1$ corresponding to the source and target domain pseudo-labels. The following proposition shows that under the assumption of an optimal domain classifier $\hat{h}'$, $d_{s,t}$ achieves JS-divergence (up to a constant shift), which upper bounds the $D_{h,\mathcal{H}}^{\text{JS}}$.

**Proposition 2.** *Suppose $d_{s,t}$ follows the form of eq. D.1 and $\hat{h}$ is the optimal domain classifier which is unconstrained, then $\max_{\hat{h}'} d_{s,t} = D_{JS}(S||T) - 2 \log 2$.*

*Proof.* From the definition, we have:

$$d_{s,t}(\hat{h}', g) = \int_{\mathcal{Z}} p_s^z(z) \log \hat{h}'(z) + p_t^z(z) \log(1 - \hat{h}'(z)) dz. \tag{D.2}$$

By taking derivatives and finding the optimal $\hat{h}^*(z)$, we get : $h^*(z) = \frac{p_s^z(z)}{p_s^z(z) + p_t^z(z)}$.

By plugging $\hat{h}^*(z)$ into equation D.1, rearranging, and using the definition of the Jensen-Shanon (JS) divergence, we get the desired result. $\square$

It is worth noting that the additional negative constant $-2 \log 2$ does not affect the optimization.

### D.3 MDD FORMULATION AND $\gamma$-WEIGHTED JS DIVERGENCE

Now let us demonstrate how our $f$-DAL framework incorporates MDD naturally. Suppose $\phi^*(t) = -\frac{1}{\gamma} \log(1 - e^t)$ and $\hat{\ell}(\hat{h}(z), \hat{h}'(z)) = \log \hat{h}'(z)_{\text{argmax } \hat{h}(z)}$. We retrieve the following result as in Zhang et al. (2019):

**Proposition 3** (Zhang et al. (2019)). *Suppose $d_{s,t}$ takes the form of MDD, i.e,*

$$\gamma d_{s,t} = \gamma \mathbb{E}_{z \sim p_s^z} \log \hat{h}'(z)_{\text{argmax } \hat{h}(z)} + \mathbb{E}_{z \sim p_t^z} \hat{h}(z) \cdot \log(1 - \hat{h}'(z)_{\text{argmax } \hat{h}(z)}). \tag{D.3}$$

*With unconstrained function class $\hat{\mathcal{H}}$, the optimal $d_{s,t}$ satisfies:*

$$\max_{\hat{h}'} \gamma d_{s,t} = (\gamma + 1) \text{JS}_\gamma(p_s^z \| p_t^z) + \gamma \log \gamma - (\gamma + 1) \log(\gamma + 1), \tag{D.4}$$

*where $\text{JS}_\gamma(p_s^z \| p_t^z)$ is $\gamma$-weighted Jensen–Shannon divergence (Huszár, 2015; Nowozin et al., 2016):*

$$\text{JS}_\gamma(p_s^z \| p_t^z) = \frac{\gamma}{\gamma + 1} \text{KL}(p_s^z \| \frac{\gamma p_s^z + p_t^z}{\gamma + 1}) + \frac{1}{\gamma + 1} \text{KL}(p_t^z \| \frac{\gamma p_s^z + p_t^z}{\gamma + 1}). \tag{D.5}$$

We remark that when $\gamma = 1$, $\text{JS}_\gamma(p_s^z \| p_t^z)$ is the original Jensen–Shannon divergence. One should also note the the additional negative constant $\gamma \log \gamma - (\gamma + 1) \log(\gamma + 1)$, which attributes to the negativity of MDD, does not affect the optimization.

$\phi^*(t) = -\frac{1}{\gamma} \log(1 - e^t)$ can be considered by rescaling the $\phi^*$ for the usual JS divergence (see Table 4). In general we can rescale $\phi^*$ for any $f$-divergence with the following lemma:

**Lemma 4** (Boyd & Vandenberghe (2004)). *For any $\lambda > 0$, the Fenchel conjugate of $\lambda \phi$ is $(\lambda \phi)^*(t) = \lambda \phi^*(t/\lambda)$, with $\text{dom}(\lambda \phi)^* = \lambda \text{dom } \phi^*$.*

# E  A TOY EXAMPLE FOR THE TRAINING DYNAMICS

Suppose that the source dataset S and target dataset T contain one sample each and are formulated: $S = \{(0.5, 1)\}$ and $T = \{0.55\}$. Let the feature extractor to be quadratic functions, and we choose linear predictors, i.e.:

$$g(x) = w_1 x^2 + x, \ \hat{h}(x) = w_2 x, \ \hat{h}'(x) = \sigma(w_3 x). \tag{E.1}$$

Let us consider the regression task with the JS divergence used to compare S and T.

$$\min_{w_1, w_2 \in \mathbb{R}} \max_{w_3 \in \mathbb{R}} \mathbb{E}_{x \sim P_s} (f_s(x) - \hat{h}(g(x)))^2 + \mathbb{E}_{x \sim P_s} \log \hat{h}'(g(x)) +$$

$$+ \mathbb{E}_{x \sim P_t} \log(1 - \hat{h}'(g(x))).$$

If we consider that $g(0.5) = g(0.55)$, $\hat{h}(g(0.5)) = 1$ and $w_3^* = 0$, then the optimal solution $(w_1^*, w_2^*, w_3^*)$ satisfies the assumption in Theorem 5. We plot the trajectories of GDA and AExG in Figure 3 and show that AExG can accelerate the convergence to the optimal solution.

