# OpenReview forum: "f-Domain-Adversarial Learning: Theory and Algorithms for Unsupervised Domain Adaptation with Neural Networks"
_ICLR.cc/2021/Conference — Reject_

### Official Review · AnonReviewer3 · 2020-10-28
**Experiments and writing need to be significantly improved.**

**Rating:** 5
**Confidence:** 4

**Review:**

Update after reading authors' response.

The authors didn't address my question on the statistical significance of their results compared with baselines.
The authors can address this question by "perform a double-sided student-t test to verify whether the performance difference between your method and the baselines are statistically significant."

I suggest the authors to perform experiments to compare with SOTA baselines in the entire field of domain adaptation, instead of just comparing with a subset of algorithms that are based on adversarial training. After all, you want to see how your work stands in the entire field, rather than limited to a sub-field.

------------------------

This paper studies unsupervised domain adaptation. The authors derive a generalization bound that utilizes a new measure of discrepancy between distributions based on a variational characterization of f-divergences. They develop an algorithm for domain-adversarial learning for the family of f-divergences.

Overall, I recommend to reject this paper, due to the following major concerns: 1) writing is not self-contained; 2) important baselines are missing; 3) insufficient justification of the advantage of the proposed method; 4) experimental results are not strong.

My major concerns of this paper include:
1. The writing needs to be significantly improved, especially the experiment section. For example,  it is very difficult to find out what those baselines in Table 2 refer to. The paper needs to be self-contained. Directing the readers to a third paper for important details such as experimental settings and results is not proper.

2. Some state-of-the-art baselines are not compared with. For example, Contrastive Adaptation Network for Unsupervised Domain Adaptation. Guoliang Kang, Lu Jiang, Yi Yang, Alexander G Hauptmann. CVPR 2019.

3. In the generalization bound in Theorem 2, the authors utilize a new discrepancy to compare the two marginal distributions. The authors didn't articulate the advantage of this new discrepancy over existing ones. Why is it significant so that we need to study it?

4. In table 5, are the results significantly different? It would be nice to perform a double-sided student-t test to verify whether the performance difference between your method and the baselines are statistically significant.

5. Values of hyperparameters are missing, making this paper difficult to reproduce. If space is an issue, the authors can put the hyperparameters in the appendix.

6. The related works section in the appendix needs to significantly improved. There are a lot of works in the space of adversarial domain adaptation and in the broader space of general domain adaptation methods. The authors need to give a more comprehensive review. Again, the review needs to be self-contained in this paper instead of directing the readers to read other review papers. And the review needs to summarize the high-level key ideas, contributions, limitations of those papers instead of focusing on detailed math.

However, this paper does have a few strong points.

1. The theoretical analysis in Section 3 is sound. I read the proofs, which appear to be correct.

2. Section 3 and 4 are well-organized and easy to follow.

Other comments
1. What does ADAA in Table 2 refer to?

2. It's better to use an algorithm box to outline the optimization algorithm in Section 4.1.

3. The paragraph title "Experimental results" in Section 5 is confusing. This paragraph is more like an introduction of this section instead of one presenting results.

---

> ### Author Response · Authors · 2020-11-19
> **Response R3 Part 1/2**
>
> We thank the reviewer for the detailed explanation provided.
> We also thank the reviewer for finding the theoretical analysis of Section 3 sound.
>
> R-1 1) Writing is not self-contained .... directing the readers to a third paper for important details such as experimental settings and results is not proper.
>
> This was an oversight. We have addressed this in the revision.  We will also provide source code.
>
> R-2  Insufficient justification for the advantage of the proposed method…..  In the generalization bound in Theorem 2, the authors utilize a new discrepancy to compare the two marginal distributions. The authors didn't articulate the advantage of this new discrepancy over existing ones. Why is it significant so that we need to study it?
>
> We respectfully disagree. The motivation of the new discrepancy is stated clear from the abstract. The goal of the new discrepancy measure is to bridge the gap between theory and the state-of-the-art methods (those that rely on adversarial learning) while providing a unified framework. Our theoretical framework provides a unifying view that connects theory and practical algorithms. Importantly, it can explain previous theoretical results and previous algorithms under the same framework. This was not possible before.
> Most importantly,  the insights from our theory allow us to derive a new adversarial algorithm that is both conceptually and practically different from previously analyzed adversarial approaches (i.e those that follow [1]).  Our resulting algorithmic approach(f-DAL) is  different from other adversarial adaptation frameworks because our theory shows that we should use a per-category domain classifier instead of a “global” domain-classifier or “discriminator” (please notice eq 4.3 \hat H is the same).(page 7  middle paragraphs ).
>
> This is explained in the abstract, introduction ( paragraphs : 3, 4 and 5). First paragraph of 3.1 and last paragraph of 3.2. It is also restated in the conclusions.
>
> R-3 In table 5, are the results significantly different?
>
> We do not have Table 5 in the paper, we assume the reviewer refers to Table 3.
> We respectfully disagree with the view that the performance of our model is not very significant. Indeed, we believe it is significant  that by connecting theory and algorithms ( and thus correcting previous adversarial domain adaptation methods that follows DANN [1]), we can  achieve performances that previous methods can only achieve through adversarial learning + additional techniques (i.e regularization, conditioning, and tweaking of extra hyperparams per dataset).  Our goal was only to show that fixing this key ingredient (inspired by the theory) is powerful enough to achieve better results than  other SoTA.

---

> > ### Author Response · Authors · 2020-11-19
> > **Response R3 Part 2/2**
> >
> > R -4 Some state-of-the-art baselines are not compared with. For example, Contrastive Adaptation Network for Unsupervised Domain Adaptation[1].
> >
> > We thank the reviewer for the reference. In the comparison vs SoTA methods, we focus on the comparison  with SoTAs that rely on domain-adversarial training since this is the focus of our work. Notice this is not the case for [1]. Moreover, the “Contrastive Adaptation Method” mostly relies on additional techniques  that neither our method nor the proposed baselines used (i.e clustering  in order to compute target pseudo-labels, class-aware sampling and alternative optimization).  Furthermore, closer inspection to Table 3 in [1]  shows that when any of the aforementioned techniques is not used they do not outperform f-DAL or the proposed baselines (i.e MDD w/o AO).These techniques further introduce complexity and computational overhead.
> > We believe our results can be further improved with these additional techniques and further tuning. However, this problem is left for further research as it does not constitute the essence of our work.
> > Our goal is to propose a unifying framework that connects the theory used to explain domain-adversarial training i.e DANN (Ganin et al., 2016) (and similar SoTA algorithms), and the algorithms themselves. The new theory results in a new adversarial framework (Sec 4), which outperforms previous SoTA adversarial methods (Table 2 and Table 3), but notice this was not our goal.
> >
> > We have clarified this in our updated version  of the manuscript and also referenced the Contrastive Adaptation framework.
> >
> > R-5 The related works section in the appendix needs to be significantly improved.
> >
> > We did not provide a related work section for two reasons 1) because we clearly summarize the high-level key ideas, contributions and limitations of previous work (as the reviewer suggested through our work). For example, Introduction (i.e paragraphs : 3 , 4  and 5). Last paragraph of 3.2.  2) due to space constraints. Thus, the related work in the appendices aims to point the reader to a more extensive literature, and it is added for completeness.
> >
> > R-6 Values of hyperparameters are missing, making this paper difficult to reproduce.
> > We have included these in the revised manuscript, this was an oversight. We will also release source code for reproducibility.
> >
> > R-7 The paragraph title "Experimental results" in Section 5 is confusing.  This paragraph is more like an introduction of this section instead of one presenting results.
> > This has been fixed in the revised version.
> >
> > [1] Contrastive Adaptation Network for Unsupervised Domain Adaptation arvix1901.00976

---

> > > ### Comment · AnonReviewer3 · 2020-11-25
> > > **compare with SOTA baselines**
> > >
> > > I suggest the authors to perform experiments to compare with SOTA baselines in the entire field of domain adaptation, instead of just comparing with a subset of algorithms that are based on adversarial training. After all, you want to see how your work stands in the entire field, rather than limited to a sub-field.

---

> > ### Comment · AnonReviewer3 · 2020-11-25
> > **Statistical significance of results**
> >
> > The authors didn't address my question on the statistical significance of their results compared with baselines. The authors can address this question by "perform a double-sided student-t test to verify whether the performance difference between your method and the baselines are statistically significant."

---

### Official Review · AnonReviewer1 · 2020-10-28
**Review #1**

**Rating:** 4
**Confidence:** 5

**Review:**

--------After rebuttal (updated)---------------

Thanks for your detailed response. I would like to apologize for my late feedback.
Since your rebuttal is long without proper organization (5 pages without properly using markdown), I may miss several points. I directly commented my feedback on my original review, `by the marked text`

Several of my concerns have been surely fixed and the understanding of the proposed approach is much more clear. However, it is still difficult to change my score because of the following reasons:
1. *Current paper still requires careful polishing for facilitating reading.*  e.g. The authors claimed several times they will update the paper with pseudo-code, it is still missing after discussion. The paper and the rebuttal are still dense, which make the reader
difficult to understand.

2. *The diverse datasets (such as NLP, digits experiment in the original DANN paper) and additional in-depth empirical analysis (not only accuracy) are indeed necessary.*
I think you do not need to claim a significantly better performance with SOTA. The detailed empirical analysis is more important.

I hope my additional comments and feedback can help you improve your paper.

 ------------------------------------------



Summary:

This paper proposed a hypothesis based f-divergence domain adaptation theory and algorithm. They found previous popular divergences such as H-divergence and MDD can be viewed as the special cases of f-divergence. Finally, they validated their practical benefits on office-31 and office-home dataset.

------------------------------------------------------

Overall review

Pros:

[1] New DA theory on f-divergence.

[2] The proof is technically sound.

Cons:

[1] The significance of the paper (theoretical and practical) is rather unclear.

[2] The experimental results are weak.

[3] Some details need better justifications and discussions.

Based on these, I recommend a rejection but encourage a major revision for resubmission.

-----------------------------------------------------------

Detailed explanations

[A] Significance. (theoretical and practical aspects)

Theoretical aspects

[1] Using f-divergence in DA is not new. For instance, [1][2][3][4] have already discussed DA by using $\beta$ (Rényi)-divergence, $\chi$ divergence. I do not understand why such a **unified** f-divergence does not include these divergences. I guess because the theory requires Lipschitz f-divergence, but this is too limited. A theory on general f-divergence (not only Lipschitz $\phi^{\star}$) is highly expected.

`Partially fixed. Missing point: it is possible to prove f-divergence only with f(1)=0 and f is convex? (the original definition of f-divergence)`


[2] The theoretical assumptions are too restrictive for practice. I just list some of them:

f-divergence is Lipschitz,
$\ell\in[0,1]$ and strong triangle inequality,
deterministic label function setting.

In contrast, previous work [2,3] has proposed strong theoretical guarantees for the cross-entropy loss and **stochastic data generation process** $P(y|x)$ instead of $f(x)$. Given this paper is in the same nature as f-divergence. More general settings are **highly** expected.

`Not fixed. My concern is that the previous approach has proved stochastic settings in some f-divergence. This should be properly discussed and addressed.`


[3] The contribution of the optimization part is unclear. If this part is the theoretical contribution, an optimization convergence bound should be provided.

`Partially fixed. The term *rigorous* is confusing. You did not provide a bound how to define it is rigorous.`


[4] The theoretical assumptions on representation learning are strong. I am not sure how these are realistic in the office-31 and office-home dataset. The conditional distribution is not clearly defined. More discussions on this part are highly expected.

`Not fixed. I am still not sure how these are realistic. The conditional distribution is referred as a labeling function on $x$ f(x)? or $z$ f(z), or probability distribution P(y|x) or P(y|z). This is quite important in the representation learning approach.`


[5] Theorem 3 seems too coarse in deep learning. (Since this paper claims they have a strong theoretical contribution in deep learning. I think this ought to be addressed).

`Fixed`


[6] The KL-divergence is **not tight** if we use dual terms of f-divergence. A Donsker-Varadhan Theorem based theoretical analysis on KL divergence is expected.


  `Partially fixed. The Wasserstein distance makes me more confused about introducing f-divergence. Since JS, TV has such problems, why not Wasserstein distance. It is always better.`


Practical aspects.

[7] From a practical aspect, f-divergence adversarial training is not new.

Based on the f-gans paper, we can practically easily replace JS divergence with any-other f-divergence without any technical difficulty. For example, [5] derives some DA algorithms on general f-divergence. From this perspective, the new empirical insights are rather limited.

`Fixed. Thanks for your additional figures. I understand your new practice.`


[8]  The extra gradient approach presented in the experiments is rather unclear and inspired by existing optimization papers. I think a clear and complete discussion is expected. The current version seems like a plug-in approach.

`Not fixed. Maybe the rebuttal is too dense. I can not get your point`


[9] The whole empirical parts are presented in a dense mode, I suggest some parts can be safely moved to the appendix.

`Partially fixed. It is better but the dense wrap figure makes it still hard to read.`


[10] It is Ok not to provide the code, but a detailed algorithm description or protocol ought to be provided. This is particularly important when this paper aims at proposing several new ideas.

`Not fixed. This is really important. Author claimed they will write a description but i have not seen even in the appendix`


[B] Empirical results and analysis

[1] From all the results. The empirical gain is too limited. Besides, the compared baselines are limited or not recent SOTA. (The newest results only come from ICML 2019).

`Not fixed. If you claim your approach is SOTA, at least a statistical test should be added to show it is indeed significant.`


[2] In office-home. I do not know why the std values are not reported in this case. The other 7 tasks are missing in the paper and appendix. The number of baselines is significantly fewer than office-31.

`Partially fixed. STD still not reported and still fewer baselines. The author claimed previous papers did not report these..But do you think it is a good thing not reporting variance? Particularly in office-31 you reported std, which is really odd.`


[3] The standard digits datasets are not evaluated. Since the Digits dataset is trained from scratch and different from the pre-training approach. Testing on these is expected.

`It seems to be ignored..I think this is essential to validate the theory.`


[4] When testing KL-divergence, the dual term of KL-divergence is not tight. I think a Donsker-Varadhan Theorem based practice should be tested.

`Fixed.`


[C] Other details

The only analysis of this paper is the numerical accuracy of office-home/31 and toy data. A deeper analysis of why the proposed f-divergence is better than the previous is quite lacking. I can not feel the strong motivation of why preferring f-divergence.
I suggest to put additional analysis such as T-SNE, the optimal value on **real-data** (such as p_t(z)/p_s(z)), the convergence behavior, the evolution of f-divergence, ablation study.

`Not fixed. The only indicator in this paper is numerical accuracy. I still do not know what makes f-divergence better.`


--------------------------------------------
Suggestions

I suggest a major revision on the improved theory and empirical analysis (not simply accuracy) on the benefits of f-divergence.


Ref:

[1] Multiple source adaptation and the Rényi divergence. UAI 2009

[2] A new PAC-Bayesian perspective on domain adaptation. ICML 2016

[3] Algorithms and theory for multiple-source adaptation. NeurIPS 2018

[4] Revisiting (\epsilon, \gamma, \tau)-similarity learning for domain adaptation. NeurIPS 2018

[5] Domain adaptation with asymmetrically-relaxed distribution alignment. ICML 2019

---

> ### Author Response · Authors · 2020-11-19
> **Response R1 Part 1/3**
>
> We thank the reviewer for the detailed explanation provided.
> We also thank the reviewer  for taking the time to verify our proofs and finding them technically sound.
>
> R-1.   Using f-divergence in DA is not new. For instance, [1][2][3][4]  have already discussed DA by using (Rényi)-divergence and \chi^2 divergence,… ”
>
> Our novelty is in providing a unifying framework that connects theory and practical algorithms through its variational characterization. We can explain previous theoretical results and previous practical algorithms (based on JS) under the same framework. This follows from our analysis.
>
> Notice that methods that follow [6] (which constitutes most of SoTA in Unsupervised Domain Adaptation (UDA)) are explained with insights from the theory of Ben-David et al 2010 or similar reductions of the TV. In practice, the divergence that they minimize is JS (Appendix D). We claim novelty for resolving this disconnect, and additionally for showing how a proper adversarial framework (different from those that follow [6]) results from resolving such a disconnect.
>
> Regarding \chi^2 divergences, we did use Pearson \chi^2 and Neyman chi^2 (Table 1 and Table 4). Indeed, we found the Pearson \chi^2  works best across all datasets and scenarios in our experimental results.
>
> We do not consider (Rényi)-divergences because these measures are not exactly f-divergences, they are obtained as monotone transformation of an appropriate f-divergence, “as stated in 7.12 paragraph 1 second sentence in [9]”. Our focus is on f-divergences which are already very general.
>
> That said, the \beta_q divergence in [2] is compatible with our framework because according to [2], eq. (7): \beta_q(S||T)^q = \int p_T (p_S/p_T)^q dx.  This is a f-divergence with \phi(x) = x^q with q>0 and phi* is C^1 in [a,b] then Lipschitz.
>
> For additional details  and quantitative results about the theoretical and algorithmic significance please check “Theoretical and Algorithmic Significance “ in the General Response.
>
>
> R-2  “A theory on general f-divergence (not only Lipschitz ) is highly expected….. The theoretical assumptions are too restrictive for practice
>
> We do not completely understand why the restriction of  \phi^* to be Lipschitz in a compact domain ( i.e [a,b] in the paper) is too restrictive for practical applications.
> If \phi* is continuously differentiable in [a,b] then the Lipschitz assumption holds. This assumption holds for most f-divergences which to the best of our knowledge are used in most practical applications. Notice that, the domain of \phi^* can be shifted by taking phi(x) to \phi(x) + c x, or scaled with Lemma 4, which changes the f-divergence Def. 1 only up to a shift of constant or up to a rescaling. In this way, \phi* can always include the domain [0, 1] as required by Theorem 2 and 3.  Regarding \chi^2 divergences, we did use Pearson \chi^2 and Neyman chi^2 (Table 1 and Table 4).
>
> Please let us know what other proper f-divergences (beyond the ones stated in Table 1 and Table 4) are used in practical applications for domain adaptation with neural networks and do not satisfy these properties.
>
> R-3 -> “deterministic label function setting.”... and “stochastic data generation process”
>
> We agree with this but we believe this should be left for further work as it does not constitute the essence of the current project.  In this work, we aim to connect the previous theoretical works used to explain DANN [6] and similar SoTA algorithms, and the algorithms themselves. This  results in  a new domain adversarial framework as explained above.
>
> R-4 The contribution of the optimization part is unclear. If this part is the theoretical contribution, an optimization convergence bound should be provided.
>
> The optimization part shows that  the desired solution of f-DAL is a Stackelberg Equilibrium. To the best of our knowledge, we are the first to rigorously show this. This key insight allows us to motivate the use of more suitable optimizers for this problem. We  believe that providing a convergence bound  for the optimization objective of f-DAL should be left for further work. This is a highly non-convex minimax optimization problem and it might be  impossible to prove some results  unless strong assumptions are stated. We emphasize the key idea of this section is  to show that the desired solution of the proposed algorithmic framework is under mild assumptions a Stackelberg equilibrium. This is stated in the introductory paragraph of 4.1.
> We have updated the section name to Optimality in f-DAL to make this clearer.

---

> > ### Author Response · Authors · 2020-11-19
> > **Response R1 Part 2/3**
> >
> > R-5  The theoretical assumptions on representation learning are strong. I am not sure how these are realistic in the office-31 and office-home dataset.
> >
> > As stated in the paper, we completely agree with the fact that they might be  strong assumptions. However, it is not completely clear whether this holds or not in practice.  Most of the SoTA works in real datasets on UDA rely on learning adversarial invariant representation in a way or another, and thus this is assumed (sometimes not explicitly stated). Most of these works (including ours) are able to achieve high transfer performance (in practice).
> >
> > R-6 -The conditional distribution is not clearly defined.
> > This was an oversight and has been corrected.
> >
> > R-7 Practical aspects [7] From a practical aspect, f-divergence adversarial training is not new …. we can practically easily replace JS divergence with any-other f-divergence without any technical difficulty. For example, [5] derives some DA algorithms on general f-divergence. From this perspective, the new empirical insights are rather limited.
> >
> > We respectfully disagree. We are not merely replacing previous frameworks (i.e those that follow [6]) with other divergence as it was done in [5] or without technical difficulty as the reviewer is pointing.
> > Inspired by our theory, and the proposed divergence measure, we correct previous adversarial frameworks (those that follow DANN [6] and are explained with insights from Ben-David et al 2010)  by using a per-category domain classifier instead of a global domain classifier used as a “discriminator”. This is both conceptually and practically different from the previous methods studied. In those, there is a “global” domain classifier mapping Z->R “a discriminator”. This is not the case for us.  Our auxiliary classifier can be also interpreted intuitively. It determines whether the “predicted category” is from the source or the target domain.  This is highly desired in practice  since practitioners do not have to think about finding an architecture for “a discriminator”.
> > For additional details  and quantitative results about the theoretical and algorithmic significance please check “Theoretical and Algorithmic Significance “ in the General Response.
> >
> > R-8 “The extra gradient approach presented in the experiments is rather unclear and inspired by existing optimization”
> >
> > We do not claim novelty for the extra-gradient approach. We claim novelty for showing that under mild assumptions the desired solution of f-DAL is  a Stackelberg equilibrium. This key observation inspired the use of optimizers that have been shown to be more appropriate for problems where the desired solution is a Stackelberg equilibrium (i.e extragradient methods). The experiments corroborate this is true as suggested by the theory.
> >
> > R-10 “It is Ok not to provide the code, but a detailed algorithm description or protocol ought to be provided. “
> >
> > We did not provide an algorithm box because the algorithm itself follows from [7] and there are page limit constraints. We will also release code upon acceptance which we believe would make reproducibility easier. We will write the pseudocode in the appendix in an updated revision.
> >
> > R-11  Empirical results and analysis . From all the results. The empirical gain is too limited.
> >
> > We respectfully disagree with this view. We believe it is impressive that by connecting theory and algorithm and fixing traditional adversarial approaches as explained above, we can  achieve performances that previous methods can only achieve through adversarial learning + additional techniques (i.e regularization, conditioning, and tweaking of extra hyperparams per dataset). Our goal was only to show that correcting this key detail (inspired by the theory) is powerful enough to achieve better results than  other SoTA.  We argue that a fair comparison would be vs DANN[1] using JS as the example provided in the general response. Additionally, notice that MDD is part of our framework and requires additional hyperpameter tuning for each dataset ( Table 4 in  [8])

---

> > > ### Author Response · Authors · 2020-11-19
> > > **Response R1 Part 3/3**
> > >
> > > R-12 In office-home. I do not know why the std values are not reported in this case. The other 7 tasks are missing
> > >
> > > We did not provide std because it was not reported in previous sources (i.e Table 2 in [8], Table 3 in [9] so we do not know std values for baselines. The other 7 tasks were missing because of space constraints, this is corrected in the 9 pages revision.
> > >
> > > R-13 “The dual term of KL-divergence is not tight. I think a Donsker-Varadhan Theorem based practice should be tested.”
> > >
> > > We acknowledge the famous Donsker-Varadhan representation of the KL divergence and we appreciate the reviewer for pointing it out. However, this formulation is specific for KL divergence and cannot be generalized to other f-divergences, while our Theorem 2 is general. In our work, we aim for a general framework that connects theory and algorithm instead of introducing new ad-hoc optimization objectives.
> > > Moreover, it is well-known that KL can grow very fast when the supports of the two measures differ a lot (see e.g. the WGAN paper [10]), compared to JS. Therefore, according to our Thm 2, KL may not give the tightest bound. Even with DV theorem there is no strong theoretical guarantee.
> > >
> > > [1] Multiple source adaptation and the Rényi divergence. UAI 2009
> > >
> > > [2] A new PAC-Bayesian perspective on domain adaptation. ICML 2016
> > >
> > > [3] Algorithms and theory for multiple-source adaptation. NeurIPS 2018
> > >
> > > [4] Revisiting (\epsilon, \gamma, \tau)-similarity learning for domain adaptation. NeurIPS 2018
> > >
> > > [5] Domain adaptation with asymmetrically-relaxed distribution alignment. ICML 2019
> > >
> > > [6] Domain-Adversarial Training of Neural Networks, JMLR
> > >
> > > [7] A Variational Inequality Perspective on Generative Adversarial Networks arxiv1802.10551
> > >
> > > [8] Bridging Theory and Algorithm for Domain Adaptation arxiv1904.05801
> > >
> > > [9] Yury Polyanskiy, Information Theoretic Methods in Statistics and Computer Science, MIT 2019-2020 Available at: http://people.lids.mit.edu/yp/homepage/data/LN_fdiv.pdf
> > >
> > > [10]  Wasserstein GAN, arxiv1701.07875

---

> > > > ### Comment · AnonReviewer1 · 2020-11-23
> > > > **Thanks for your response**
> > > >
> > > > Dear authors,
> > > >
> > > > Thanks for your detailed rebuttal.
> > > > I have updated my reviews and feedbacks. I hope this can help you improve the quality of your paper.

---

> ### Author Response · Authors · 2020-11-23
> **Post-Rebuttal Response**
>
> We appreciate and thank your feedback.  Below our response.
>
> `
> >[A1] Partially fixed. Missing point: it is possible to prove f-divergence only with f(1)=0 and f is convex? (the original definition of f-divergence)
>
> We apologize but we do not understand this point. We derive a framework for the complete family of f-divergences (Definition 1, Csizar 1967). We additionally extend the framework for shifted and scaled f-divergences to include even more cases (i.e relaxing f(1)=0). Last paragraph section 6.  In Tables 1 and 4, we provide popular f-divergences.
> Assuming the reviewer refers to the domain of \phi^* or Lipschitzness on Theorem 3. We explain below in Response R1-Part 1 R-2 why this is not a limitation.
>
> >[A2] Not fixed. My concern is that the previous approach has proved stochastic settings in some f-divergence.
>
> We agree but this is not the goal of our work.
>
> >[A6] Partially fixed. The Wasserstein distance makes me more confused about introducing f-divergence.....
>
> We do not deal with Wasserstein distance in our work. IPMs are a different family of divergences. The use of Wasserstein distances brings practical challenges (i.e 1-Lipschitness), and thus most practical algorithms in domain adversarial learning do not rely on them. This is why we prefer f-divergences. We aim for a framework that bridges theory and algorithms.
>
> >[A3] and [A8]  The contribution of the optimization part is unclear. Extragradient.
>
> >[A3] Partially fixed. The term *rigorous* is confusing. You did not provide a bound how to define it is rigorous.
>
> >[A8] Not fixed. Maybe the rebuttal is too dense.
>
> We do not provide convergence bound because we do not introduce a new optimizer. The goal of this section is to show that optimality in this framework is a particular type of equilibrium and thus raise awareness about the need for more suitable optimizers in domain-adversarial learning (we use the extra-gradient to show a practical example). We mention rigorous because we prove the existence of the equilibrium (optimality) theorem 5 and theorem 6. Appendix C.
>
> >[A4] The theoretical assumptions on representation learning are strong.
> > Not fixed. I am still not sure how these are realistic. The conditional distribution is referred to as a labeling function on $x$ f(x)?
>
> We do not claim they are realistic. We claim assumptions are needed for DA. Otherwise, it is not possible to solve. Domain-Adversarial methods rely on this assumption in a way or another. P_s(y|x) = P_t(y|x) page 5 footnote explained with words, also referenced is provided.
>
> >[A10] It is Ok not to provide the code........
>
> >Not Fixed.
>
> We will provide source code and pseudocode in a final version.
>
> >[A2] The digits datasets and additional in-depth empirical analysis are necessary.
>
> We provide large-scale experiments in real datasets that mimics real scenarios. Pretraining is a common practice these days. . We agree digit datasets are a nice to have   toy example. However, performance and insights obtained on these sometimes do not correlate with real-world scenarios. This is why we prioritize large scale experiments.  We believe not including experiments on MNIST-like datasets should not be a major reason for scoring.

---

### Official Review · AnonReviewer4 · 2020-10-29
**the theoretical results are not so novel as claimed in the paper**

**Rating:** 5
**Confidence:** 5

**Review:**

###############

Summary: This paper proposes a new generalization bound for domain adaptation based on f-divergences. Accordingly, a new algorithmic framework is also derived.

###############

Pros:

Some theoretical results are new – to the best of my knowledge, there are no existing works establishing learning bounds for DA with f-divergences.
One benefit of f-divergence is that it can offer a very general framework that accommodates various discrepancy measures for DA. The authors empirically evaluate different choices of divergences over several benchmarks.
The paper is well-written and easy to follow.

Cons:

My main concern is the theoretical contribution and motivation of this work. Given a large variety of divergence measures (e.g., H-divergence, JS-divergence, Wasserstein distance, MDD…), this paper does not give me any new theoretical insights compared to previous results. From an algorithmic perspective, extending JS divergence to more general f-divergences has already studied in GAN training [1].

In addition, most of the theoretical analysis follows standard steps of existing works. For example, the high-level idea of Definition 3 follows the notion of Disparity Discrepancy in [2] and is also similar to the notion of source-guided discrepancy in [3]. The proofs of Lemma 1, Lemma 2, Theorem 2, and Theorem 3 are either straightforward or follow standard techniques in DA (e.g., [2,4,5]).

The empirical results in Table 3 indicate that the improvement of f-DAL is marginal compared to other algorithms (e.g., MDD). Furthermore, the improvement may even come from AExG, rather than f-DAL itself. I would speculate that f-DAL can even be outperformed by other baseline algorithms without AExG.

################

Overall: While I appreciate the f-divergences generalization bounds derived in this paper, I didn’t get any new perspective after reading the paper. Every point of the theoretical results in this paper seems familiar to me.

[1] Sebastian Nowozin, Botond Cseke, and Ryota Tomioka. f-gan: Training generative neural samplers using variational divergence minimization. In D. D. Lee, M. Sugiyama, U. V. Luxburg, I. Guyon, and R. Garnett (eds.), Advances in Neural Information Processing Systems 29, pp. 271–279. Curran Associates, Inc., 2016.

[2] Yuchen Zhang, Tianle Liu, Mingsheng Long, and Michael Jordan. Bridging theory and algorithm for domain adaptation. In Kamalika Chaudhuri and Ruslan Salakhutdinov (eds.), Proceedings of the 36th International Conference on Machine Learning, volume 97 of Proceedings of Machine Learning Research, pp. 7404–7413, Long Beach, California, USA, 09–15 Jun 2019.

[3] Kuroki, S., Charonenphakdee, N., Bao, H., Honda, J., Sato, I., and Sugiyama, M. Unsupervised domain adaptation based on source-guided discrepancy. In AAAI Conference on Artificial Intelligence (AAAI), 2019.

[4] Shai Ben-David, John Blitzer, Koby Crammer, Alex Kulesza, Fernando Pereira, and Jennifer Wortman Vaughan. A theory of learning from different domains. Machine learning, 79(1-2):151–175, 2010.

[5] Han Zhao, Remi Tachet Des Combes, Kun Zhang, and Geoffrey Gordon. On learning invariant representations for domain adaptation. In International Conference on Machine Learning, pp. 7523-7532, 2019.

---

> ### Author Response · Authors · 2020-11-19
> **Response R4 Part 1/2**
>
> We thank the reviewer for the feedback and acknowledging the non-existence of works establishing learning bounds for DA with f-divergences.
>
> R-1 - “My main concern is the theoretical contribution and motivation of this work. Given a large variety of divergence measures (e.g., H-divergence, JS-divergence, Wasserstein distance, MDD…), this paper does not give me any new theoretical insights compared to previous results.”
>
> a)
> The large variety of divergence measures used in the theoretical analysis of DA and the disconnect between these and the ones used in practice for adversarial learning (mostly JS)  is one of the strong motivations of our work. Our theoretical framework provides a unifying view that connects theory and practical algorithms. Importantly, our approach explains previous theoretical results and previous algorithms under the same framework (as the reviewer pointed).
> This allows us to explain how some optimization algorithms used in practice are less appropriate than others.
>
> Most importantly,  the insights from our theory allows us to derive a new adversarial algorithm that is both conceptually and practically different from previously analyzed adversarial approaches (i.e those that follow [1]).  Our resulting algorithmic approach is  different from others' adversarial adaptation frameworks because our theory shows that we should use a per-category domain classifier instead of a “global” domain-classifier or “discriminator” (please notice eq 4.3 \hat H is the same). This was also explained in words on page 6 first paragraph (middle of page 7 in updated revision and marked in red) .  Please notice that this key insight makes the algorithm both conceptually and practically different from previously analyzed adversarial approaches (i.e those that follow [1]).
>
> We have further emphasized this in the revised manuscript, and provided a diagram that illustrates this important and key contribution.
>
>
>
> Quantitative Significance of this key contribution.
>
> Please check “Theoretical and Algorithmic Significance” in the general response.
>
>
>
> R-2 “From an algorithmic perspective, extending JS divergence to more general f-divergences has already studied in GAN training[1]”
>
> We respectfully disagree with this view. Our goal is conceptually and practically different to the one presented in f-GAN, and so it is the training algorithm. In addition to having a discrepancy term, our objective also incorporates a supervised loss (unlike f-GAN). The intuition of a discriminator is also different (as explained above). Conceptually, we aim to learn features that are indistinguishable from the point of view of the domains but discriminative enough for a classifier to make accurate predictions. GANs are generative models. Thus, while in both cases the variational characterization of f-divergences is exploited, the goal is different. From an algorithmic point of view, we also show optimality in our framework is at the Stackelberg Equilibrium (Sect 4 and Appendix C). This motivates the use of more appropriate optimizers (i.e ExtraGradient algorithms).  Please check “Theoretical and Algorithmic Significance” in the general response for more details in the difference vs our method and traditional domain adversarial approaches.
>
> R-3 “Most of the theoretical analysis follows standard steps of existing works.”
>
> We do not see this as a limitation. Indeed, this shows the need for a framework that can summarize and explain all those works and beyond (with a focus on what works in practice).  The key observation is that most of them belong to the same “f-divergence” family and thus can be generalized through a reduction of its variational characterization. Thus, we believe the simplicity of the analysis has nothing to do with its significance and is actually one of its advantages.
>
>
> R-4 “The empirical results in Table 3 indicate that the improvement of f-DAL is marginal compared to other algorithms (e.g., MDD)”
>
> We respectfully disagree with this view. We believe it is significant that by connecting theory and algorithm and fixing traditional adversarial approaches as explained above (R-1 a) ), we can  achieve performances that previous methods can only achieve through additional techniques (i.e regularization, conditioning, and tweaking of extra hyperparams per dataset). We argue that a fair comparison would be vs DANN [1] using JS as the example provided above (please check general response "Theoretical and Algorithmic Significance" ).
>
> Most importantly, concerning the “marginal improvement”, MDD is part of our framework (gamma JS in Appendix D) but requires additional hyperpameter tuning per dataset (see Table 4 in  [2]). MDD is competitive with our approach only after this additional hyperparameter tuning that our approach does not use (nor require).
> The goal of the result tables was to show that we can obtain state-of-the-art results with less tuning.

---

> > ### Author Response · Authors · 2020-11-19
> > **Response R4 Part 2/2**
> >
> > R-5  The improvement may even come from AExG, rather than f-DAL itself.
> >
> > The use of the AExG in f-DAL is motivated in our framework after noticing and proving that optimality for the general framework is a Stackelberg Equilibrium (Sect 4 and Appendix C). We believe this  is a new and interesting theoretical result with practical applications . As such, we consider AExG as part of f-DAL algorithm itself.
> > Please see the answer below for more details.
> >
> > R-6 “Table 3…” “I would speculate that f-DAL can even be outperformed by other baseline algorithms without AExG.”
> >
> > This speculation is not correct. We show this in the following experiment.
> > Office-Home (Table3)
> >
> > Avg Performance CDAN 65.8
> >
> > Avg Performance MDD/ (or f-DAL gamma JS (Appendix D))  (SGD): 68.1
> >
> > Avg Performance fDAL(Pearson, SGD): 68.3
> >
> > Avg Performance fDAL(Pearson, AExG): 68.7
> >
> > AExG indeed improves slightly the performance and provides faster convergence, which is inline with the insights from theoretical results presented in Sect 4 and Appendix C, and its motivation. However, it is not necessary (if convergence speed is not an issue).  We emphasize MDD is a special case of f-DAL with gamma-weighted JS divergence.
> >
> > We added this experiment and analysis to the revised version.
> >
> > R-7 ”Every point of the theoretical results in this paper seems familiar to me”
> >
> > Shouldn’t this be seen as an advantage of our study that it follows naturally from existing theory? We believe simplicity in the analysis should not be seen as a limitation. Even if it is simple, it needs to be written and proved.
> >
> > References:
> >
> > [1] Domain-Adversarial Training of Neural Nets ,JMLR 2016
> >
> > [2] Bridging Theory and Algorithm for Domain Adaptation ICML 2019 arxiv1904.05801

---

> > > ### Comment · AnonReviewer4 · 2020-11-24
> > > **Thanks for your response**
> > >
> > > Sorry for the delayed response. While I appreciate the response which clarifies some points, some of my concerns are still not addressed.
> > >
> > > I understand the unifying view is the main contribution of this work. But it is still not clear to me what is the benefit of it given the previous works that already built the connections between theories and algorithms. More importantly, the authors failed to address my concern regarding the empirical evaluation, which was also raised by other reviewers. Specifically, I disagree with the argument that a fair comparison would be vs DANN [1] using JS as the example provided in the paper. If the authors claim that the SoTA is achieved through additional techniques, they should provide concrete empirical results to verify it (e.g., f-DAL with these techniques).
> > >
> > > I increase my score to 5 given the clarification of the per-category domain classifier, which I think is novel from an algorithmic perspective. But I still don’t think the current work is ready for publishing in ICLR.

---

> > > > ### Author Response · Authors · 2020-11-24
> > > > **Response Post-Rebuttal**
> > > >
> > > > Thanks for your feedback and response and for also acknowledging the novel algorithmic perspective.
> > > >
> > > > > I understand the unifying view is the main contribution of this work. But it is still not clear to me what is the benefit of it given the previous works that already built the connections between theories and algorithms.
> > > >
> > > > One of the main benefits is that it leads to a new algorithmic framework(f-DAL)  which the reviewer seems to also appreciate as novel after clarification (i.e a per-category classifier). We arrive at it through the unifying theoretical framework. The second is that our work explains previous theoretical and practical results with a single notion of divergence, which is the one used in practice,  which the reviewer also acknowledged as a novel theoretical result.
> > > >
> > > > >The authors failed to address my concern regarding the empirical evaluation, which was also raised by other reviewers.
> > > >
> > > > We show experimentally in the rebuttal (Response R4 Part 2) that the improvement vs previous SoTA is not because of the optimizer as the reviewer mentioned. We also show MDD is f-DAL (gamma-JS Appendix D).
> > > >
> > > > > If the authors claim that the SoTA is achieved through additional techniques, they should provide concrete empirical results to verify it (e.g., f-DAL with these techniques).
> > > >
> > > > We claim f-DAL as presented achieves SoTA vs other domain adversarial methods. This is verified and shown experimentally in Table 2 and Table 3. We said f-DAL with additional techniques may achieve better results. That said, f-DAL is already SoTA vs adversarial methods as claimed, please look at Table2 and Table 3.
> > > >
> > > > >I increase my score to 5 given the clarification of the per-category domain classifier, which I think is novel from an algorithmic perspective.
> > > >
> > > > Thank you, this is one of the main benefits of connecting theory and algorithms into a unifying framework.  And it is one of the main benefits vs previous work.

---

### Official Review · AnonReviewer2 · 2020-10-30
**A paper with a good theory but  weak experimental evaluation**

**Rating:** 5
**Confidence:** 4

**Review:**

Further comments after the rebuttal

First, I would thank the authors for the great efforts in trying to address my comments. I should say that many of my previous concerns have been clarified. Again, I like the theoretical part. Nonetheless, I agreed with some other reviewers that the experiments seemed not to fully convince me.  Particularly to myself, the authors may want to show some analysis on how their method could truly stand out by even a toy example, which may further enhance the paper. I tend to keep my original rating  after reading all the rebuttal messages in the whole thread.

====================
This paper proposed a f-domain adversarial learning framework (f-DAL) using the complete family of f-divergences as domain discrepancy measurements. The proposed method extended the seminal works of Ben-David et al. (2007; 2010a;b); Mansour et al. (2009) that provided generalization bounds for UDA based on a special type of f-divergence. To enable the complete family of f-divergences to measure the domain distribution discrepancy, the variational characterization of f-divergences is leveraged to estimates f-divergences from samples by turning the estimation problem into variational optimization. Furthermore, a new type of discrepancy was used to compare two marginal distribution and its corresponding generalization bound was tailored for a general class of f-divergence, which can mitigate two limitations of the seminal works. In addition, the optimal solution of f-DAL is a Stackelberg equilibrium, which allows f-DAL to incorporate the latest optimizers from the game-optimization literature, such as Aggressive Extra-Gradient (AExG).

The key strength of this paper is, it enables a generalized version of f-divergences that can be used for adversarial domain adaptation. This is valuable for UDA algorithms in many application areas. As verified in the experiments, the proposed method achieved comparable results in Office-31 and Office Home dataset.

The paper was well organized and written. However, there are some major concerns as follows:
(1)	Although the authors provided theoretical insight for their method, the performance improvements are not very significant.

(2)	The ablation study needs to be further conducted. It is necessary to eliminate the Aggressive Extra-Gradient (AExG) and other learning strategies, such as spectral normalization (SN) and GRL warm-up strategy when compared with state-of-the-art methods. More specifically, the authors employed an integrated model that makes the comparison experiments seemingly unfair especially in the case of using MDD as the baseline. Although achieving the Stackelberg equilibria is a good characteristic of f-DAL, AExG may also increase the performance. Furthermore, the spectral normalization (SN) and GRL warm-up strategy may also potentially improve the performance. However, these techniques were seemingly not applied in other comparison methods in Table 2 and Table 3.

(3)	The experiments may not answer the general question ‘(2) Is there a better universal notion of f-divergence that achieves significant performance gains across different datasets?’. In Figure 3, it is hard to say that there is a consistent increment tendency for a specific f-divergence across different transfer tasks. More importantly, it is better to demonstrate if choosing different f-divergence can lead to consistent performance increase on more diverse datasets or applications.

(4)	For a typical UDA theory method, it is better to show results on the VisDA-2017 dataset.

(5)	The authors offered some hints to generalize the proposed method to a multi-class scenario. It is better to provide theoretical insight/details.

(6)	It may enhance the paper if more illustrative or even toy examples can be conducted to further show clearly the advantages of the proposed method. Again, the improvement may not be very obvious as observed in the experiments though it is theoretically interesting.

---

> ### Author Response · Authors · 2020-11-19
> **Response R2 Part 1/2**
>
> We thank the reviewer for the feedback. We address the questions below.
>
> R-1)  “ Although the authors provided theoretical insight for their method, the performance improvements are not very significant.”
>
> We believe it is significant  that by connecting theory and algorithms (and thus correcting previous adversarial domain adaptation methods that follows DANN [1]), we can achieve performance that previous methods can only achieve through adversarial learning + additional techniques (i.e regularization, conditioning, and tweaking of extra hyperparams per dataset). In practical terms, the connection between theory and algorithms means replacing a domain classifier with a per-category domain classifier (as the theory suggests) and using a particular divergence from the family (also inspired by the theory).  Thus, we respectfully disagree with the view that the performance of our model is not very significant.
>
> To be more precise, our method requires one hyper-parameter less than MDD. This hyperparameter needs to be determined per dataset ( i.e Table 4 in [1] and section 5.1 therein), we do not have that limitation.   We additionally emphasize that MDD is part of our framework itself (it is a gamma-weight JS divergence Appendix D.3), and that the gamma extra hyperparameter can be motivated for any divergence (Sect 4). We decided not to use it as it requires additional tuning. In comparison with CDAN (the other SoTA algorithm), our method does not use a conditional discriminator or entropy minimization (which requires additional hyperparameters as well) and still manages to outperform it.
> As already mentioned, our framework is general in the sense that it includes a larger family of divergences (i.e. the family of f-divergences) and the general formulation of adversarial adversarial training as a  zero-sum game leads to better optimization.
>
> Notice that the additional  techniques  can be applied on top of our framework to improve the performance further (i.e the per-category domain classifier can be conditioned, and regularization terms can be added to the loss function).
> Overall, we believe that the aim for simplicity should not be confused with the performance strength of our framework. Our goal was to propose a unifying framework for different works in the literature, our results can be further improved with additional tuning.
>
> In 2b), we address questions related to fairness in the experiment set-up.
>
> R-2)   “The ablation study needs to be further conducted. It is necessary to eliminate the Aggressive Extra-Gradient (AExG) and other learning strategies”
>
> a) The ablation study that is asked in order to eliminate the effects of the Aggressive Extra-Gradient (AExG) vs the conventional setting (SGD) was shown in Figure 4 (top) (Figure 5 left side in the updated revision).
> Notice that this was performed for every divergence, for every pair of datasets and using the  same training condition that our experiments in Table 2. This is all explained in the section “What do we get by the “extra” gradient?”.  Inspection of  that figure shows that for the Pearson X^2 (i.e the result shown in Table 2 vs SoTA baselines), the AExG led to a relative improvement of less than 0.5 in AVG (Specifically the improvement is +0.35).
> Notice that AExG is not the reason for our method outperforming the baselines. The use of the AExG in f-DAL is motivated in our framework after noticing and proving that optimality for the general framework is a Stackelberg Equilibrium (Sect 4 and Appendix C). As mentioned, AExG indeed improves performance and faster convergence, which is inline with the insights from theoretical results presented in Sect 4 and Appendix C but it is not the reason for our method outperforming baselines.
>
> Due to the 8 page format, we believe that our experimental section was too compressed in the first submitted version, we extended and clarified it in the revised 9 page version.

---

> > ### Author Response · Authors · 2020-11-19
> > **Response R2 Part 2/2**
> >
> > R-3) “spectral normalization (SN) and GRL warm-up strategy may also potentially improve the performance. However, these techniques were seemingly not applied in other comparison methods in Table 2 and Table 3.“
> >
> > b) We respectfully disagree with the view that our experimental setup is unfair in comparison with the state-of-the-art methods due to the use of the GRL-warmup and SN. Specifically, the reviewer argues that this has not been applied in other comparison methods.
> > The use of warm up in the Gradient Reversal Layer (GRL) has been exploited in domain adversarial scenarios starting with the initial paper that introduces the GRL eq 14 in [2].
> > The two best performing  state-of-the-art baselines in our comparison (i.e CDAN and MDD) both use a similar strategy.  The reviewer can check page 7 paragraph before Table 1 for CDAN [4]. To see this in MDD, we refer the reviewer to the official implementation at this address [3]
> >
> >
> > With respect to the use of Spectral Normalization (SN), we did not see any practical improvement by using it. This is only applied to the classifier head (i.e the last two fully connected layers) and the reason is to avoid gradient issues and instabilities during training for some divergences in the first epochs.
> >
> > We emphasize more these points in the revised submission. We also provide additional details of the training setup (which follows MDD, CDAN) in the revised version.
> >
> >
> > R-4  “The experiments may not answer the general question ‘(2) Is there a better universal notion of f-divergence that achieves significant performance gains across different datasets?’” … “ it is hard to say that there is a consistent increment tendency for a specific f-divergence across different transfer tasks.”
> >
> > a) We respectfully disagree with this point. In Figure 4 (previously figure 3), we compare 5 popular f-divergences among 6 tasks (pair of datasets) with  31 categories.  Each experiment was run with three different seeds, so effectively Figure 4 shows the result of 5*6*3=90 experiments. The backbone of these networks is Resnet-50.  In 5 out of the 6 scenarios, the Pearson X^2 performs better. This is also the case on average. Among all of the tasks/experiments, both the JS and the  Pearson perform better than the rest. Thus, choosing either JS or Pearson led in our experiment to consistent performance increase.  We explain this in Section 5 under paragraph “Comparing f-divergences”. In the same section, we additionally provide empirical insights into how choosing a particular divergence affects the transfer performance.  For convenience, we summarize here the average among all the task (These results are obtained from figure 4 last column)
> >
> > Avg-Resnet-50 (baseline)  75.3
> >
> > Avg - KL    84.5
> >
> > Avg - KL-rev  86.5
> >
> > Avg - TV 86.9
> >
> > Avg - JS  89.2
> >
> > AVG-Pearson 89.5
> >
> > R-5 A paper with a good theory but weak experimental evaluation
> >
> > In Figure 4, we compare 5 popular f-divergences among 6 tasks (pair of datasets) with  31 categories (Office-31). Three seeds are run for experiment and the average is reported. Thus, this figure is the result of  5*6*3=90 experiments with backbone network being Resnet-50.
> > Additionally, we run two optimizers(SGD, AExG) for 5 popular f-divergences. This  among 6 tasks (pair of datasets).   Three seeds are also run for experiment and avg is reported. This is all reported in Figure 5 left.  Quantitatively this means 2*5*6*3=180 experiments.
> >
> > We additionally run experiments with SGD and AExG for the Pearson X2 divergence in Office-Home dataset. Thus 12*3*2= 72 experiments.
> >
> > Overall, our experimental section summarizes results of training 342 neural networks in large scale datasets and real worlds networks (Resnet) .  Due to the 8 page format, we believe that our experimental section was too compressed in the first submitted version, we extended and clarified it in the revised 9 pages version.
> >
> > [1] Bridging Theory and Algorithm for Domain Adaptation ICML 2019  arxiv1904.05801
> >
> > [2] Unsupervised Domain Adaptation by Backpropagation  ICML 2015
> >
> > [3] https://github.com/thuml/MDD/blob/master/model/MDD.py#L21-L24
> >
> > [4] Conditional Adversarial Domain Adaptation, NeurIPS 2018 arxiv1705.10667

---

### Author Response · Authors · 2020-11-19
**General Response**

We thank the reviewers for their feedback. We are happy that R1, R4 acknowledge that there are no existing works establishing learning bounds for DA with f-divergences, and finding our work novel. We also thank R1 and R3  for taking the time to verify our proofs and finding them technically sound. Similarly, we are glad that R2 appreciated the value of our theory.

We have updated our paper in order to address the reviewers’ feedback. We have also added proper clarifications to highly important points that the reviewers might have overseen, and to better explain the contributions. For convenience, modified parts in the updated manuscript are marked with red color.

---

> ### Author Response · Authors · 2020-11-19
> **General Key Points**
>
> Below we summarize a few general key points:
>
> 1- Theoretical and Algorithmic Significance
>
> (The theory not only provides a unifying framework, it also leads to a new algorithmic framework. This  is different from previous methods both conceptually and practically)
>
> Our work provides a unifying view that connects theory and practical algorithms. Importantly, our approach explains previous theoretical results and previous algorithms under the same framework (as some of the reviewers recognized). We believe unifying different approaches into a general framework is a valuable contribution. Additionally, this allows us to explain how some optimization algorithms used in practice are less appropriate than others.
>
> Most importantly, the insights from our theory allows us to derive a new adversarial algorithm that is both conceptually and practically different from previously analyzed domain adversarial approaches (i.e those that follow [1]).  Our theory shows that we should use a per-category domain classifier instead of a “global” domain-classifier or “discriminator” (please notice eq 4.3 \hat H is the same, also  Definition 3).  Please notice that this key insight makes the algorithm different from previously analyzed adversarial approaches (i.e those that follow [1]).  In previous works, there is a “global” domain classifier mapping Z->R used as “a discriminator”. This is not the case for us.
>
> We have further emphasized this in the revised manuscript and provided a diagram that illustrates this important and key contribution.
>
> This difference can be experimentally quantified by comparing f-DAL (JS) (per-category domain classifier)  vs DANN [1] (“global” domain classifier and JS). To avoid any possible misunderstanding we compare both using SGD(GDA).
>
> DANN (JS) (SGD ) [1]  Avg Performance.  82.2
>
> Ours f-DAL (JS) (SGD )  Avg Performance: 89.2
>
> We believe the difference in performance is significant and comes directly out of the theoretical insights obtained from our framework.
>
> We have added the results from DANN previously in Table 2  to Figure 4 to make it clearer.
>
>
> 2- Contribution of the optimization section
>
> (The optimization part is about interpreting optimality in our framework, thus showing there are more suitable optimizers.)
>
> We emphasize the key idea of the optimization section is to show that the desired solution of the proposed algorithmic framework is under mild assumptions a Stackelberg equilibrium. We believe this is a new and interesting theoretical result with practical applications. For example, it motivates the use of more suitable optimizers. To the best of our knowledge, we are the first to rigorously show this in a domain adaptation framework.
>
> We have updated the section name to Optimality in f-DAL and clarified this in the manuscript.
>
> Experimentally, the use of more suitable optimizers (i.e AExG) improves slightly the performance and leads to faster convergence, which is inline with the insights from theoretical results presented in Section 4 and Appendix C. That said, our method achieves comparable good results using SGD (GDA).
>
> We emphasized this in the manuscript by adding experiments with SGD(GDA) in Table 2 and Table 3.
>
>
> 3- Comparison vs SoTA methods and performance.
>
>
> (It is significant that with extreme simplicity and without tuning we can achieve SoTA performance. But this was not our goal  )
>
> We believe it is significant  that by connecting theory and algorithms (and thus correcting previous adversarial domain adaptation methods that follows DANN [1]), we can  achieve performance that previous methods can only achieve through adversarial learning + additional techniques (i.e regularization, conditioning, and tweaking of extra hyperparams per dataset).  Our goal is to propose a unifying framework  that connects the theory used to explain DANN [1] (and similar SoTA algorithms), and the algorithms themselves. The new theory  results in  a new adversarial framework (Sec 4).  Noticeably, our framework outperforms previous SoTA without additional tuning or techniques.  Our results can be further improved with additional tuning or techniques (i.e CDAN) since most of SoTAs either follow from [1] or are part of our framework itself  (i.e MDD = \gamma-JS). This does not constitute our focus and it is thus  deferred to future work. We compare with SoTA that rely on adversarial learning as this is the focus of our work.
>
> We have added a section in the manuscript clarifying this point.
>
> References:
>
> [1] Domain-Adversarial Training of Neural Networks, JMLR 2016

---

### Decision · Program_Chairs · 2021-01-07
**Final Decision**

**Decision:**

Reject

**Comment:**

This paper presents a novel theoretical analysis for unsupervised domain adaptation based on f-divergences. The reviews unanimously pointed out the interest and the quality of the theoretical part. However, some limitations in the experiments, presentation and the significance of the result have been raised. The authors provided a rebuttal that addresses some concerns.
However, the reviewers agree that the experimental part still requires some extension to fully support the claim of the paper, as well as some writing improvement.
The paper was evaluated to be not ready for ICLR, thus I recommend rejection.